# Dynamically Pruned Message Passing Networks for Large-scale Knowledge Graph Reasoning

**Xiaoran Xu**[1], **Wei Feng**[1], **Yunsheng Jiang**[1], **Xiaohui Xie**[1], **Zhiqing Sun**[2], **Zhi-Hong Deng**[3]
[1]Hulu, {`xiaoran.xu, wei.feng, yunsheng.jiang, xiaohui.xie`}@hulu.com
[2]Carnegie Mellon University, `zhiqings@andrew.cmu.edu`
[3]Peking University, `zhdeng@pku.edu.cn`

## Abstract

We propose *Dynamically Pruned Message Passing Networks* (DPMPN) for large-scale knowledge graph reasoning. In contrast to existing models, embedding-based or path-based, we learn an input-dependent subgraph to explicitly model reasoning process. Subgraphs are dynamically constructed and expanded by applying graphical attention mechanism conditioned on input queries. In this way, we not only construct graph-structured explanations but also enable message passing designed in Graph Neural Networks (GNNs) to scale with graph sizes. We take the inspiration from the consciousness prior proposed by Bengio (2017) and develop a two-GNN framework to encode input-agnostic full structure representation and learn input-dependent local one coordinated by an attention module. Experiments show the reasoning capability of our model to provide clear graphical explanations as well as predict results accurately, outperforming most state-of-the-art methods in knowledge base completion tasks.

## 1 Introduction

Modern deep learning systems should bring in explicit reasoning modeling to complement their black-box models, where reasoning takes a step-by-step form about organizing facts to yield new knowledge and finally draw a conclusion. Particularly, we rely on graph-structured representation to model reasoning by manipulating nodes and edges where semantic entities or relations can be explicitly represented (Battaglia et al., 2018). Here, we choose knowledge graph scenarios to study reasoning where semantics have been defined on nodes and edges. For example, in knowledge base completion tasks, each edge is represented by a triple $\langle head, rel, tail \rangle$ that contains two entities and their relation. The goal is to predict which entity might be a tail given query $\langle head, rel, ? \rangle$.

Existing models can be categorized into embedding-based and path-based model families. The embedding-based (Bordes et al., 2013; Sun et al., 2018; Lacroix et al., 2018) often achieves a high score by fitting data using various neural network techniques but lacks interpretability. The path-based (Xiong et al., 2017; Das et al., 2018; Shen et al., 2018; Wang, 2018) attempts to construct an explanatory path to model an iterative decision-making process using reinforcement learning and recurrent networks. A question is: can we construct structured explanations other than a path to better explain reasoning in graph context. To this end, we propose to learn a dynamically induced subgraph which starts with a head node and ends with a predicted tail node as shown in Figure 1.

**Graph reasoning can be powered by Graph Neural Networks.** Graph reasoning needs to learn about entities, relations, and their composing rules to manipulate structured knowledge and produce structured explanations. Graph Neural Networks (GNNs) provide such structured computation and also inherit powerful data-fitting capacity from deep neural networks (Scarselli et al., 2009; Battaglia et al., 2018). Specifically, GNNs follow a neighborhood aggregation scheme to recursively aggregate information from neighbors to update node states. After $T$ iterations, each node can carry structure information from its $T$-hop neighborhood (Gilmer et al., 2017; Xu et al., 2018a).

**GNNs need graphical attention expression to interpret.** Neighborhood attention operation is a popular way to implement attention mechanism on graphs (Velickovic et al., 2018; Hoshen, 2017) by

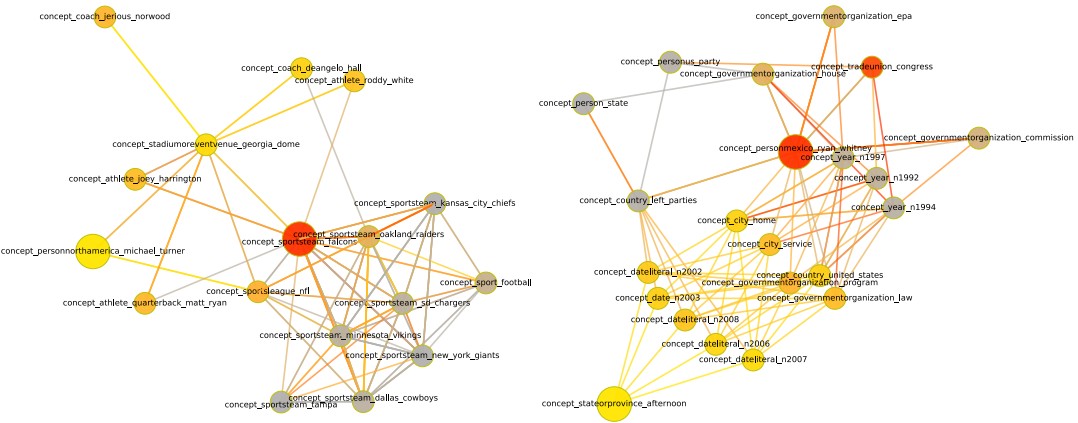

(a) The *AthletePlaysForTeam* task.

(b) The *OrganizationHiredPerson* task.

Figure 1: Subgraph visualization on two examples from NELL995's test data. Each task has ten thousands of nodes and edges. The big yellow represents a given head and the big red represents a predicted tail. Color indicates attention gained along $T$-step reasoning. Yellow means more attention during early steps while red means more attention at the end. Grey means less attention.

focusing on specific interactions with neighbors. Here, we propose a new graphical attention mechanism not only for computation but also for interpretation. We present three considerations when constructing attention-induced subgraphs: (1) given a subgraph, we first attend within it to select a few nodes and then attend over those nodes' neighborhood for next expansion; (2) we propagate attention across steps to capture long-term dependency; (3) our attention mechanism models reasoning process explicitly through pipeline disentangled from underlying representation computing.

**GNNs need input-dependent pruning to scale.** GNNs are notorious for their poor scalability. Consider one message passing iteration on a graph with $|V|$ nodes and $|E|$ edges. Even if the graph is sparse, the complexity of $\mathcal{O}(|E|)$ is still problematic on large graphs with millions of nodes and edges. Besides, mini-batch based training with batch size $B$ and high dimensions $D$ would lead to $\mathcal{O}(BD|E|)$ making things worse. However, we can avoid this situation by learning input-dependent pruning to run computation on dynamical graphs, as an input query often triggers a small fraction of the entire graph so that it is wasteful to perform computation over the full graph for each input.

**Cognitive intuition of the consciousness prior.** Bengio (2017) brought the notion of attentive awareness from cognitive science into deep learning in his *consciousness prior* proposal. He pointed out a process of disentangling high-level factors from full underlying representation to form a low-dimensional combination through attention mechanism. He proposed to use two recurrent neural networks (RNNs) to encode two types of state: unconscious state represented by a high-dimensional vector before attention and conscious state by a derived low-dimensional vector after attention.

We use two GNNs instead to encode such states on nodes. We construct input-dependent subgraphs to run message passing efficiently, and also run full message passing over the entire graph to acquire features beyond a local view constrained by subgraphs. We apply attention mechanism between the two GNNs, where the bottom runs before attention, called ***Inattentive GNN* (IGNN)**, and the above runs on each attention-induced subgraph, called ***Attentive GNN* (AGNN)**. IGNN provides representation computed on the full graph for AGNN. AGNN reinforces representation within a cohesive group of nodes to produce sharp semantics. Experimental results show that our model attains very competitive scores on HITS@1,3 and the mean reciprocal rank (MRR) compared to the best embedding-based method so far. More importantly, we provide explanations while they do not.

## 2 ADDRESSING THE SCALE-UP PROBLEM

**Notation.** We denote training data by $\{(x_i, y_i)\}_{i=1}^N$. We denote a full graph by $\mathcal{G} = \langle \mathcal{V}, \mathcal{E} \rangle$ with relations $\mathcal{R}$ and an input-dependent subgraph by $G(x) = \langle V_{G(x)}, E_{G(x)} \rangle$ which is an induced subgraph of $G$. We denote boundary of a graph by $\partial G$ where $V_{\partial G} = N(V_G) - V_G$ and $N(V_G)$ means neighbors of nodes in $V_G$. We also denote high-order boundaries such as $\partial^2 G$ where $V_{\partial^2 G} = N(N(V_G)) \cup N(V_G) - V_G$. Trainable parameters include node embeddings $\{e_v\}_{v \in \mathcal{V}}$,

relation embeddings $\{e_r\}_{r \in \mathcal{R}}$, and weights used in two GNNs and an attention module. When performing full or pruned message passing, node and relation embeddings will be indexed according to the operated graph, denoted by $\theta_{\mathcal{G}}$ or $\theta_{G(x)}$. For IGNN, we use $\mathcal{H}^t$ of size $|\mathcal{V}| \times D$ to denote node hidden states at step $t$; for AGNN, we use $H^t(x)$ of size $|V_{G(x)}| \times D$ to denote. The objective is written as $\sum_{i=1}^N l(x_i, y_i; \theta_{G(x_i)}, \theta_{\mathcal{G}})$, where $G(x_i)$ is dynamically constructed.

**The scale-up problem in GNNs.** First, we write the full message passing in IGNN as

$$\mathcal{H}^t = f_{\text{IGNN}}(\mathcal{H}^{t-1}; \theta_{\mathcal{G}}), \tag{1}$$

where $f_{\text{IGNN}}$ represents all involved operations in one message passing iteration over $\mathcal{G}$, including: (1) computing messages along each edge with the complexity[1] of $\mathcal{O}(BD|\mathcal{E}|)$, (2) aggregating messages received at each node with $\mathcal{O}(BD|\mathcal{E}|)$, and (3) updating node states with $\mathcal{O}(BD|\mathcal{V}|)$. For $T$-step propagation, the per-batch complexity is $\mathcal{O}(BDT(|\mathcal{E}| + |\mathcal{V}|))$. Considering that backpropagation requires intermediate computation results to be saved during one pass, this complexity counts for both time and space. However, since IGNN is input-agnostic, node representations can be shared across inputs in one batch so that we can remove $B$ to get $\mathcal{O}(DT(|\mathcal{E}| + |\mathcal{V}|))$. If we use a sampled edge set $\hat{\mathcal{E}}$ from $\mathcal{E}$ such that $|\hat{\mathcal{E}}| \approx k|\mathcal{V}|$, the complexity can be further reduced to $\mathcal{O}(DT|\mathcal{V}|)$.

The pruned message passing in AGNN can be written as

$$H^t(x) = f_{\text{AGNN}}(H^{t-1}(x), \mathcal{H}^t; \theta_{G(x)}). \tag{2}$$

Its complexity can be computed similarly as above. However, we cannot remove $B$. Fortunately, subgraph $G(x)$ is not $\mathcal{G}$. If we let $x$ be a node $v$, $G(x)$ grows from a single node, i.e., $G^0(x) = \{v\}$, and expands itself each step, leading to a sequence of $(G^0(x), G^1(x), \ldots, G^T(x))$. Here, we describe the expansion behavior as *consecutive expansion*, which means no jumping across neighborhood allowed, so that we can ensure that

$$G^t(x) \subseteq G^{t-1}(x) \cup \partial G^{t-1}(x) \subseteq G^{t-2}(x) \cup \partial^2 G^{t-2}(x). \tag{3}$$

Many real-world graphs follow the *small-world* pattern, and the *six degrees of separation* implies $G^0(x) \cup \partial^6 G^0(x) \approx \mathcal{G}$. The upper bound of $G^t(x)$ can grow exponentially in $t$, and there is no guarantee that $G^t(x)$ will not explode.

**Proposition.** *Given a graph $\mathcal{G}$ (undirected or directed in both directions), we assume the probability of the degree of an arbitrary node being less than or equal to $d$ is larger than $p$, i.e., $P(\deg(v) \leq d) > p, \forall v \in V$. Considering a sequence of consecutively expanding subgraphs $(G^0, G^1, \ldots, G^T)$, starting with $G^0 = \{v\}$, for all $t \geq 1$, we can ensure*

$$P\left(|V_{G^t}| \leq \frac{d(d-1)^t - 2}{d - 2}\right) > p^{\frac{d(d-1)^{t-1}-2}{d-2}}. \tag{4}$$

The proposition implies the guarantee of upper-bounding $|V_{G^t(x)}|$ becomes exponentially looser and weaker as $t$ gets larger even if the given assumption has a small $d$ and a large $p$ (close to 1). We define graph increment at step $t$ as $\Delta G^t(x)$ such that $G^t(x) = G^{t-1}(x) \cup \Delta G^t(x)$. To prevent $G^t(x)$ from explosion, we need to constrain $\Delta G^t(x)$.

**Sampling strategies.** A simple but effective way to handle the large scale is to do sampling.

1. $\Delta G^t(x) = \hat{\partial} G^{t-1}(x)$, where we sample nodes from the boundary of $G^{t-1}(x)$.

2. $\Delta G^t(x) = \partial \widehat{G^{t-1}(x)}$, where we take the boundary of sampled nodes from $G^{t-1}(x)$.

3. $\Delta G^t(x) = \hat{\partial} \widehat{G^{t-1}(x)}$, where we sample nodes twice from $G^{t-1}(x)$ and from $\partial \widehat{G^{t-1}(x)}$.

4. $\Delta G^t(x) = \widehat{\hat{\partial} \widehat{G^{t-1}(x)}}$, where we sample nodes three times with the last from $\hat{\partial} \widehat{G^{t-1}(x)}$.

Obviously, we have $\widehat{\hat{\partial} \widehat{G^{t-1}(x)}} \subseteq \hat{\partial} \widehat{G^{t-1}(x)} \subseteq \partial \widehat{G^{t-1}(x)}$ and $G^{t-1}(x) \cup \partial \widehat{G^{t-1}(x)} \subseteq G^{t-1}(x) \cup \partial G^{t-1}(x)$. Further, we let $N_1$ and $N_3$ be the maximum number of sampled nodes in $\partial \widehat{G^{t-1}(x)}$ and the last sampling of $\hat{\partial} \widehat{G^{t-1}(x)}$ respectively and let $N_2$ be per-node maximum sampled neighbors in $\hat{\partial} G^{t-1}(x)$, and then we can obtain much tighter guarantee as follow:

---

[1] We assume per-example per-edge per-dimension time cost as a unit time.

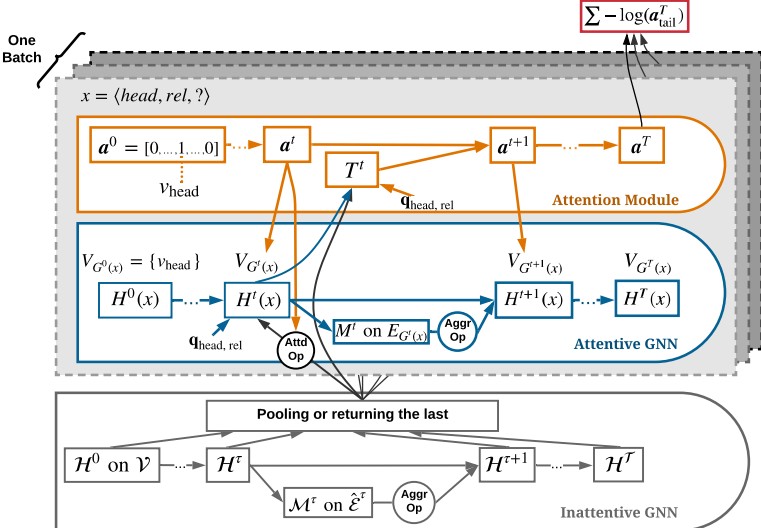

Figure 2: Model architecture used in knowledge graph reasoning.

1. $P(|V_{\Delta G^t(x)}| \le N_1(d-1)) > p^{N_1}$ for $\partial \widehat{G^{t-1}(x)}$.

2. $P(|V_{\Delta G^t(x)}| \le N_1 N_2) = 1$ and $P(|V_{\Delta G^t(x)}| \le N_1 \cdot \min(d-1, N_2)) > p^{N_1}$ for $\hat{\partial} \widehat{G^{t-1}(x)}$.

3. $P(|V_{\Delta G^t(x)}| \le \min(N_1 N_2, N_3)) = 1$ for $\overline{\hat{\partial} \widehat{G^{t-1}(x)}}$.

**Attention strategies.** Although we guarantee $|V_{G^T(x)}| \le 1 + T \min(N_1 N_2, N_3)$ by $\overline{\hat{\partial} \widehat{G^{t-1}(x)}}$ and constrain the growth of $G^{t-1}(x)$ by decreasing either $N_1 N_2$ or $N_3$, smaller sampling size means less area explored and less chance to hit target nodes. To make efficient selection rather than random sampling, we apply attention mechanism to do the top-$K$ selection where $K$ can be small. We change $\overline{\hat{\partial} \widehat{G^{t-1}(x)}}$ to $\widetilde{\hat{\partial} \widehat{G^{t-1}(x)}}$ where $\sim$ represents the operation of attending over nodes and picking the top-$K$. There are two types of attention operations, one applied to $G^{t-1}(x)$ and the other applied to $\hat{\partial} \widehat{G^{t-1}(x)}$. Note that the size of $\hat{\partial} \widehat{G^{t-1}(x)}$ might be much larger if we intend to sample more nodes with larger $N_2$ to sufficiently explore the boundary. Nevertheless, we can address this problem by using smaller dimensions to compute attention, since attention on each node is a scalar requiring a smaller capacity compared to node representation vectors computed in message passing.

## 3 DPMPN MODEL

### 3.1 ARCHITECTURE DESIGN FOR KNOWLEDGE GRAPH REASONING

Our model architecture as shown in Figure 2 consists of:

- *IGNN module*: performs full message passing to compute full-graph node representations.
- *AGNN module*: performs a batch of pruned message passing to compute input-dependent node representations which also make use of underlying representations from IGNN.
- *Attention Module*: performs a flow-style attention transition process, conditioned on node representations from both IGNN and AGNN but only affecting AGNN.

**IGNN module.** We implement it using standard message passing mechanism (Gilmer et al., 2017). If the full graph has an extremely large number of edges, we sample a subset of edges, $\hat{\mathcal{E}}^\tau \subset \mathcal{E}$, randomly each step. For a batch of input queries, we let node representations from IGNN be shared across queries, containing no batch dimension. Thus, its complexity does not scale with batch size and the saved resources can be allocated to sampling more edges. Each node $v$ has a state $\mathcal{H}_{v,:}^\tau$ at step $\tau$, where the initial $\mathcal{H}_{v,:}^0 = e_v$. Each edge $\langle v', r, v \rangle$ produces a message, denoted by $\mathcal{M}_{\langle v',r,v \rangle,:}^\tau$ at step $\tau$. The computation components include:

- Message function: $\boldsymbol{\mathcal{M}}^{\tau}_{\langle v',r,v\rangle,:} = \psi_{\text{IGNN}}(\boldsymbol{\mathcal{H}}^{\tau}_{v',:}, \boldsymbol{e}_r, \boldsymbol{\mathcal{H}}^{\tau}_{v,:})$, where $\langle v',r,v\rangle \in \hat{\mathcal{E}}^{\tau}$.

- Message aggregation: $\overline{\boldsymbol{\mathcal{M}}}^{\tau}_{v,:} = \frac{1}{\sqrt{N^{\tau}(v)}} \sum_{v',r} \boldsymbol{\mathcal{M}}^{\tau}_{\langle v',r,v\rangle,:}$, where $\langle v',r,v\rangle \in \hat{\mathcal{E}}^{\tau}$.

- Node state update function: $\boldsymbol{\mathcal{H}}^{\tau+1}_{v,:} = \boldsymbol{\mathcal{H}}^{\tau}_{v,:} + \delta_{\text{IGNN}}(\boldsymbol{\mathcal{H}}^{\tau}_{v,:}, \overline{\boldsymbol{\mathcal{M}}}^{\tau}_{v,:}, \boldsymbol{e}_v)$, where $v \in \mathcal{V}$.

We compute messages only for sampled edges, $\langle v',r,v\rangle \in \hat{\mathcal{E}}^{\tau}$, each step. Functions $\psi_{\text{IGNN}}$ and $\delta_{\text{IGNN}}$ are implemented by a two-layer MLP (using leakyReLu for the first layer and tanh for the second) with input arguments concatenated respectively. Messages are aggregated by dividing the sum by the square root of $N^{\tau}(v)$, the number of neighbors that send messages to $v$, preserving the scale of variance. We use a residual adding to update each node state instead of a GRU or a LSTM. After running for $\mathcal{T}$ steps, we output a pooling result or simply the last, denoted by $\boldsymbol{\mathcal{H}} = \boldsymbol{\mathcal{H}}^{\mathcal{T}}$, to feed into downstream modules.

**AGNN module.** AGNN is input-dependent, which means node states depend on input query $x = \langle head, rel, ?\rangle$, denoted by $\boldsymbol{H}^t_{v,:}(x)$. We implement pruned message passing, running on small subgraphs each conditioned on an input query. We leverage the sparsity and only save $\boldsymbol{H}^t_{v,:}(x)$ for visited nodes $v \in V_{G^t(x)}$. When $t = 0$, we start from node $head$ with $V_{G^0(x)} = \{v_{head}\}$. When computing messages, denoted by $\boldsymbol{M}^t_{\langle v',r,v\rangle,:}(x)$, we use an attending-sampling-attending procedure, explained in Section 3.2, to constrain the number of computed edges. The computation components include:

- Message function: $\boldsymbol{M}^t_{\langle v',r,v\rangle,:}(x) = \psi_{\text{AGNN}}(\boldsymbol{H}^t_{v',:}(x), \boldsymbol{c}_r(x), \boldsymbol{H}^t_{v,:}(x))$, where $\langle v',r,v\rangle \in E_{G^t(x)}^2$, and $\boldsymbol{c}_r(x) = [\boldsymbol{e}_r, \boldsymbol{q}_{head}, \boldsymbol{q}_{rel}]$ represents a context vector.

- Message aggregation: $\overline{\boldsymbol{M}}^t_{v,:}(x) = \frac{1}{\sqrt{N^t(v)}} \sum_{v',r} \boldsymbol{M}^t_{\langle v',r,v\rangle,:}(x)$, where $\langle v',r,v\rangle \in E_{G^t(x)}$.

- Node state attending function: $\widetilde{\boldsymbol{H}}^{t+1}_{v,:}(x) = a^{t+1}_v \boldsymbol{W} \boldsymbol{\mathcal{H}}_{v,:}$, where $a^{t+1}_v$ is an attention score.

- Node state update function: $\boldsymbol{H}^{t+1}_{v,:}(x) = \boldsymbol{H}^t_{v,:}(x) + \delta_{\text{AGNN}}(\boldsymbol{H}^t_{v,:}(x), \overline{\boldsymbol{M}}^t_{v,:}(x), \boldsymbol{c}^{t+1}_v(x))$, where $\boldsymbol{c}^{t+1}_v(x) = [\widetilde{\boldsymbol{H}}^{t+1}_{v,:}(x), \boldsymbol{q}_{head}, \boldsymbol{q}_{rel}]$ also represents a context vector.

Query context is defined by its head and relation embeddings, i.e., $\boldsymbol{q}_{head} = \boldsymbol{e}_{head}$ and $\boldsymbol{q}_{rel} = \boldsymbol{e}_{rel}$. We introduce a node state attending function to pass node representation information from IGNN to AGNN weighted by a scalar attention score $a^{t+1}_v$ and projected by a learnable matrix $\boldsymbol{W}$. We initialize $\boldsymbol{H}^0_{v,:}(x) = \boldsymbol{\mathcal{H}}_{v,:}$ for node $v \in V_{G^0(x)}$, letting unseen nodes hold zero states.

**Attention module.** Attention over $T$ steps is represented by a sequence of node probability distributions, denoted by $\boldsymbol{a}^t$ ($t = 1, 2 \ldots, T$). The initial distribution $\boldsymbol{a}^0$ is a one-hot vector with $\boldsymbol{a}^0[v_{head}] = 1$. To spread attention, we need to compute transition matrices $\boldsymbol{T}^t$ each step. Since it is conditioned on both IGNN and AGNN, we capture two types of interaction between $v'$ and $v$: $\boldsymbol{H}^t_{v',:}(x) \sim \boldsymbol{H}^t_{v,:}(x)$, and $\boldsymbol{H}^t_{v',:}(x) \sim \boldsymbol{\mathcal{H}}_{v,:}$. The former favors visited nodes, while the latter is used to attend to unseen neighboring nodes.

$$\boldsymbol{T}^t_{:,v'} = \text{softmax}_{v \in N^t(v')}\Big(\sum_r \alpha_1(\boldsymbol{H}^t_{v',:}(x), \boldsymbol{c}_r(x), \boldsymbol{H}^t_{v,:}(x)) + \alpha_2(\boldsymbol{H}^t_{v',:}(x), \boldsymbol{c}_r(x), \boldsymbol{\mathcal{H}}_{v,:})\Big)$$
$$\alpha_1(\cdot) = \text{MLP}(\boldsymbol{H}^t_{v',:}(x), \boldsymbol{c}_r(x))^{\text{T}} \boldsymbol{W}_1 \text{MLP}(\boldsymbol{H}^t_{v,:}(x), \boldsymbol{c}_r(x)) \qquad (5)$$
$$\alpha_2(\cdot) = \text{MLP}(\boldsymbol{H}^t_{v',:}(x), \boldsymbol{c}_r(x))^{\text{T}} \boldsymbol{W}_2 \text{MLP}(\boldsymbol{\mathcal{H}}_{v,:}, \boldsymbol{c}_r(x))$$

where $\boldsymbol{W}_1$ and $\boldsymbol{W}_2$ are two learnable matrices. Each MLP uses one single layer with the leakyReLu activation. To reduce the complexity for computing $\boldsymbol{T}^t$, we use nodes $v' \in V_{\widetilde{G^t(x)}}$, which contains nodes with the $k$-largest attention scores at step $t$, and use nodes $v$ sampled from $v'$'s neighbors to compute attention transition for the next step. Due to the fact that nodes $v'$ result from the top-$k$ pruning, the loss of attention may occur to diminish the total amount. Therefore, we use a renormalized version, $\boldsymbol{a}^{t+1} = \boldsymbol{T}^t \boldsymbol{a}^t / \|\boldsymbol{T}^t \boldsymbol{a}^t\|$, to compute new attention scores. We use attention scores at the final step as the probability to predict the tail node.

---

[2]In practice, we can use a smaller set of edges than $E_{G^t(x)}$ to pass messages as discussed in Section 3.2

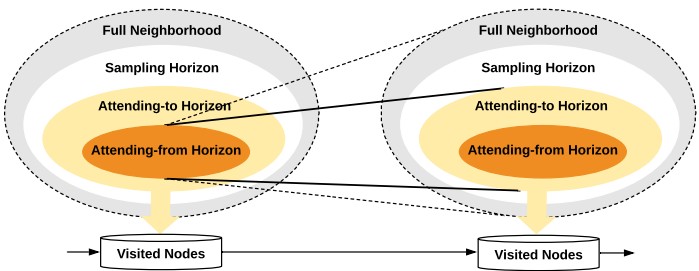

Figure 3: Iterative attending-sampling-attending procedure balancing coverage and complexity.

## 3.2 COMPLEXITY REDUCTION BY ITERATIVE ATTENDING, SAMPLING AND ATTENDING

AGNN deals with each subgraph relying on input $x$ and keeps a few selected nodes in $V_{G^t(x)}$, called *visited nodes*. Initially, $V_{G^0(x)}$ contains only one node $v_{head}$, and then $V_{G^t(x)}$ is enlarged by adding new nodes each step. When propagating messages, we can just consider the one-hop neighborhood each step. However, the expansion goes so rapidly that it covers almost all nodes after a few steps. The key to address the problem is to constrain the scope of nodes we can expand the boundary from, i.e., the core nodes which determine where we can go next. We call it the *attending-from horizon*, $\widetilde{G^t(x)}$, selected according to attention scores $\boldsymbol{a}^t$. Given this horizon, we may still need node sampling over the neighborhood $N(\widetilde{G^t(x)})$ in some cases where a hub node of extremely high degree exists to cause an extremely large neighborhood. We introduce an *attending-to horizon*, denoted by $\widehat{N}(\widetilde{G^t(x)})$, inside the *sampling horizon*, denoted by $\widehat{N}(\widetilde{G^t(x)})$. The attention module runs within the sampling horizon with smaller dimensions exchanged for sampling more neighbors for a larger coverage. In one word, we face a trade-off between coverage and complexity, and our strategy is to sample more but attend less plus using small dimensions to compute attention. We obtain the attending-to horizon according to newly computed attention scores $\boldsymbol{a}^{t+1}$. Then, message passing iteration at step $t$ in AGNN can be further constrained on edges between $\widetilde{G^t(x)}$ and $\widehat{N}(\widetilde{G^t(x)})$, a smaller set than $E_{G^t(x)}$. We illustrate this procedure in Figure 3.

## 4 EXPERIMENTS

**Datasets.** We use six large KG datasets: FB15K, FB15K-237, WN18, WN18RR, NELL995, and YAGO3-10. FB15K-237 (Toutanova & Chen, 2015) is sampled from FB15K (Bordes et al., 2013) with redundant relations removed, and WN18RR (Dettmers et al., 2018) is a subset of WN18 (Bordes et al., 2013) removing triples that cause test leakage. Thus, they are both considered more challenging. NELL995 (Xiong et al., 2017) has separate datasets for 12 query relations each corresponding to a single-query-relation KBC task. YAGO3-10 (Mahdisoltani et al., 2014) contains the largest KG with millions of edges. Their statistics are shown in Table 1. We find some statistical differences between train and validation (or test). In a KG with all training triples as its edges, a triple $(head, rel, tail)$ is considered as a multi-edge triple if the KG contains other triples that also connect $head$ and $tail$ ignoring the direction. We notice that FB15K-237 is a special case compared to the others, as there are no edges in its KG directly linking any pair of $head$ and $tail$ in validation (or test). Therefore, when using training triples as queries to train our model, given a batch, for FB15K-237, we cut off from the KG all triples connecting the head-tail pairs in the given batch, ignoring relation types and edge directions, forcing the model to learn a composite reasoning pattern rather than a single-hop pattern, and for the rest datasets, we only remove the triples of this batch and their inverse from the KG to avoid information leakage before training on this batch. This can be regarded as a hyperparameter tuning whether to force a multi-hop reasoning or not, leading to a performance boost of about $2\%$ in HITS@1 on FB15-237.

**Experimental settings.** We use the same data split protocol as in many papers (Dettmers et al., 2018; Xiong et al., 2017; Das et al., 2018). We create a KG, a directed graph, consisting of all train triples and their inverse added for each dataset except NELL995, since it already includes reciprocal relations. Besides, every node in KGs has a self-loop edge to itself. We also add inverse relations into the validation and test set to evaluate the two directions. For evaluation metrics, we use HITS@1,3,10 and the mean reciprocal rank (MRR) in the filtered setting for FB15K-237, WN18RR,

Table 1: Statistics of the six KG datasets. PME (tr) means the proportion of multi-edge triples in train; PME (va) means the proportion of multi-edge triples in validation; AL (va) means the average length of shortest paths connecting each head-tail pair in validation.

| Dataset | #Entities | #Rels | #Train | #Valid | #Test | PME (tr) | PME (va) | AL (va) |
|---------|-----------|-------|--------|--------|-------|----------|----------|---------|
| FB15K | 14,951 | 1,345 | 483,142 | 50,000 | 59,071 | 81.2% | 80.6% | 1.22 |
| FB15K-237 | 14,541 | 237 | 272,115 | 17,535 | 20,466 | 38.0% | **0%** | 2.25 |
| WN18 | 40,943 | 18 | 141,442 | 5,000 | 5,000 | 93.1% | 94.0% | 1.18 |
| WN18RR | 40,943 | 11 | 86,835 | 3,034 | 3,134 | 34.5% | 35.5% | 2.84 |
| NELL995 | 74,536 | 200 | 149,678 | 543 | 2,818 | 100% | 31.1% | 2.00 |
| YAGO3-10 | 123,188 | 37 | 1,079,040 | 5,000 | 5,000 | 56.4% | 56.0% | 1.75 |

Table 2: Comparison results on the FB15K-237 and WN18RR datasets. Results of [♠] are taken from (Nguyen et al., 2018), [♣] from (Dettmers et al., 2018), [♡] from (Shen et al., 2018), [♢] from (Sun et al., 2018), [△] from (Das et al., 2018), and [⊠] from (Lacroix et al., 2018). Some collected results only have a metric score while some including ours take the form of "mean (std)".

| Metric (%) | FB15K-237 | | | | WN18RR | | | |
|------------|-----------|------|-------|------|--------|------|-------|------|
| | H@1 | H@3 | H@10 | MRR | H@1 | H@3 | H@10 | MRR |
| TransE [♠] | - | - | 46.5 | 29.4 | - | - | 50.1 | 22.6 |
| DistMult [♣] | 15.5 | 26.3 | 41.9 | 24.1 | 39 | 44 | 49 | 43 |
| DistMult [♡] | 20.6 (.4) | 31.8 (.2) | - | 29.0 (.2) | 38.4 (.4) | 42.4 (.3) | - | 41.3 (.3) |
| ComplEx [♣] | 15.8 | 27.5 | 42.8 | 24.7 | 41 | 46 | 51 | 44 |
| ComplEx [♡] | 20.8 (.2) | 32.6 (.5) | - | 29.6 (.2) | 38.5 (.3) | 43.9 (.3) | - | 42.2 (.2) |
| ConvE [♣] | 23.7 | 35.6 | 50.1 | 32.5 | 40 | 44 | 52 | 43 |
| ConvE [♡] | 23.3 (.4) | 33.8 (.3) | - | 30.8 (.2) | 39.6 (.3) | 44.7 (.2) | - | 43.3 (.2) |
| RotatE [♢] | 24.1 | 37.5 | 53.3 | 33.8 | 42.8 | 49.2 | **57.1** | 47.6 |
| ComplEx-N3[⊠] | - | - | **56** | **37** | - | - | 57 | 48 |
| NeuralLP [♡] | 18.2 (.6) | 27.2 (.3) | - | 24.9 (.2) | 37.2 (.1) | 43.4 (.1) | - | 43.5 (.1) |
| MINERVA [♡] | 14.1 (.2) | 23.2 (.4) | - | 20.5 (.3) | 35.1 (.1) | 44.5 (.4) | - | 40.9 (.1) |
| MINERVA [△] | - | - | 45.6 | - | 41.3 | 45.6 | 51.3 | - |
| M-Walk [♡] | 16.5 (.3) | 24.3 (.2) | - | 23.2 (.2) | 41.4 (.1) | 44.5 (.2) | - | 43.7 (.1) |
| **DPMPN** | **28.6 (.1)** | **40.3 (.1)** | 53.0 (.3) | **36.9 (.1)** | **44.4 (.4)** | **49.7 (.8)** | 55.8 (.5) | **48.2 (.5)** |

FB15K, WN18, and YAGO3-10, and use the mean average precision (MAP) for NELL995's single-query-relation KBC tasks. For NELL995, we follow the same evaluation procedure as in (Xiong et al., 2017; Das et al., 2018; Shen et al., 2018), ranking the answer entities against the negative examples given in their experiments. We run our experiments using a 12G-memory GPU, TITAN X (Pascal), with Intel(R) Xeon(R) CPU E5-2670 v3 @ 2.30GHz. Our code is written in Python based on TensorFlow 2.0 and NumPy 1.16 and can be found by the link[3] below. We run three times for each hyperparameter setting per dataset to report the means and standard deviations. See hyperparameter details in the appendix.

**Baselines.** We compare our model against embedding-based approaches, including TransE (Bordes et al., 2013), TransR (Lin et al., 2015b), DistMult (Yang et al., 2015), ConvE (Dettmers et al., 2018), ComplE (Trouillon et al., 2016), HolE (Nickel et al., 2016), RotatE (Sun et al., 2018), and ComplEx-N3 (Lacroix et al., 2018), and path-based approaches that use RL methods, including DeepPath (Xiong et al., 2017), MINERVA (Das et al., 2018), and M-Walk (Shen et al., 2018), and also that uses learned neural logic, NeuralLP (Yang et al., 2017).

**Comparison results and analysis.** We report comparison on FB15K-23 and WN18RR in Table 2. Our model DPMPN significantly outperforms all the baselines in HITS@1,3 and MRR. Compared to the best baseline, we only lose a few points in HITS@10 but gain a lot in HITS@1,3. We speculate that it is the reasoning capability that helps DPMPN make a sharp prediction by exploiting graph-structured composition locally and conditionally. When a target becomes too vague to predict, reasoning may lose its advantage against embedding-based models. However, path-based baselines, with a certain ability to do reasoning, perform worse than we expect. We argue that it might be inappropriate to think of reasoning, a sequential decision process, equivalent to a sequence of nodes. The average lengths of the shortest paths between heads and tails as shown in Table 1 suggests a very short path, which makes the motivation of using a path almost useless. The reasoning pattern should be modeled in the form of dynamical local graph-structured pattern with nodes densely connected

---

[3]https://github.com/anonymousauthor123/DPMPN

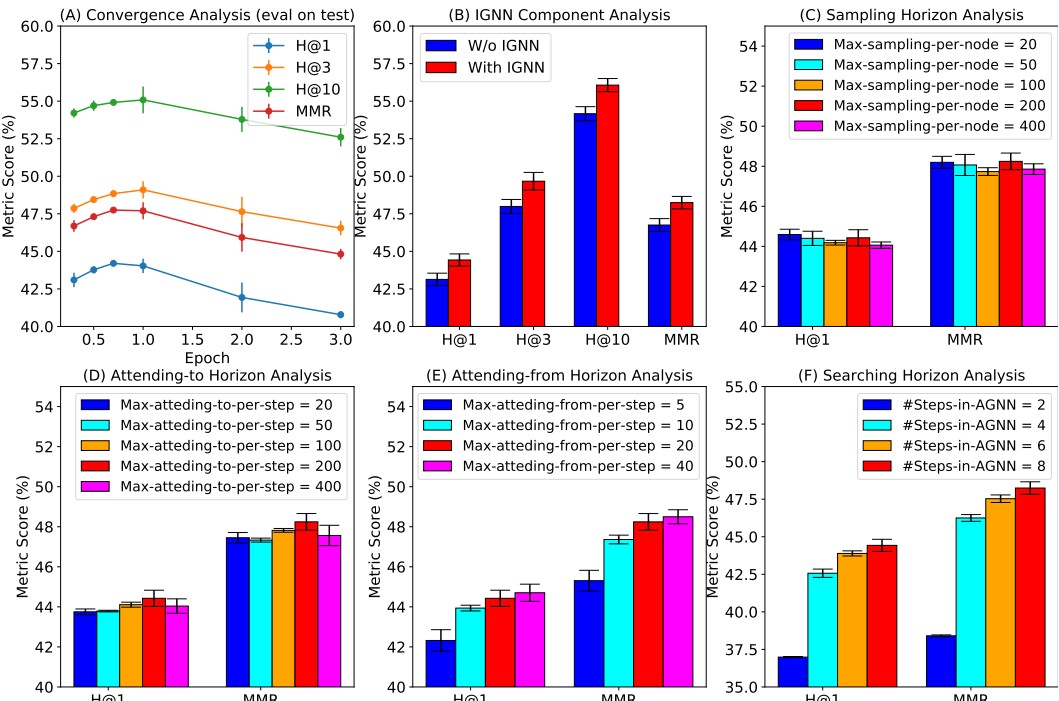

Figure 4: Experimental analysis on WN18RR. (A) Convergence analysis: we pick six model snapshots during training and evaluate them on test. (B) IGNN component analysis: *w/o IGNN* uses zero step to run message passing, while *with IGNN* uses two; (C)-(F) Sampling, attending-to, attending-from and searching horizon analysis. The charts on FB15K-237 can be found in the appendix.

with each other to produce a decision collectively. We also run our model on FB15K, WN18, and YAGO3-10, and the comparison results in the appendix show that DPMPN achieves a very competitive position against the best state of the art. We summarize the comparison on NELL995's tasks in the appendix. DPMPN performs the best on five tasks, also being competitive on the rest.

**Convergence analysis.** Our model converges very fast during training. We may use half of training queries to train model to generalize as shown in Figure 4(A). Compared to less expensive embedding-based models, our model need to traverse a number of edges when training on one input, consuming much time per batch, but it does not need to pass a second epoch, thus saving a lot of training time. The reason may be that training queries also belong to the KG's edges and some might be exploited to construct subgraphs during training on other queries.

**Component analysis.** Given the stacked GNN architecture, we want to examine how much each GNN component contributes to the performance. Since IGNN is input-agnostic, we cannot rely on its node representations only to predict a tail given an input query. However, AGNN is input-dependent, which means it can be carried out to complete the task without taking underlying node representations from IGNN. Therefore, we can arrange two sets of experiments: (1) AGNN + IGNN, and (2) AGNN-only. In AGNN-only, we do not run message passing in IGNN to compute $\mathcal{H}_{v,:}$ but instead use node embeddings as $\mathcal{H}_{v,:}$, and then we run pruned message passing in AGNN as usual. We want to be sure whether IGNN is actually useful. In this setting, we compare the first set which runs IGNN for two steps against the second one which totally shuts IGNN down. The results in Figure 4(B) (and Figure 7(B) in Appendix) show that IGNN brings an amount of gains in each metric on WN18RR (and FB15K-23), indicating that representations computed by full-graph message passing indeed help subgraph-based message passing.

**Horizon analysis.** The sampling, attending-to, attending-from and searching (i.e., propagation steps) horizons determine how large area a subgraph can expand over. These factors affect computation complexity as well as prediction performance. Intuitively, enlarging the exploring area by sampling more, attending more, and searching longer, may increase the chance of hitting a target to gain some performance. However, the experimental results in Figure 4(C)(D) show that it is not always the case. In Figure 4(E), we can see that increasing the maximum number of attending-

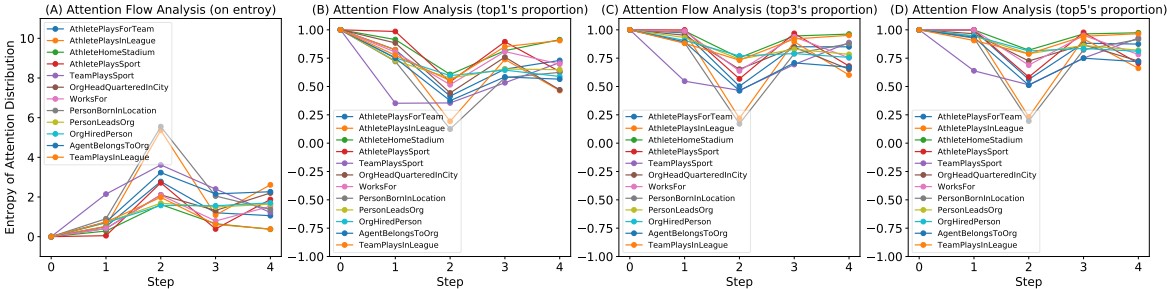

Figure 5: Analysis of attention flow on NELL995 tasks. (A) The average entropy of attention distributions changing along steps for each single-query-relation KBC task. (B)(C)(D) The changing of the proportion of attention concentrated at the top-1,3,5 nodes per step for each task.

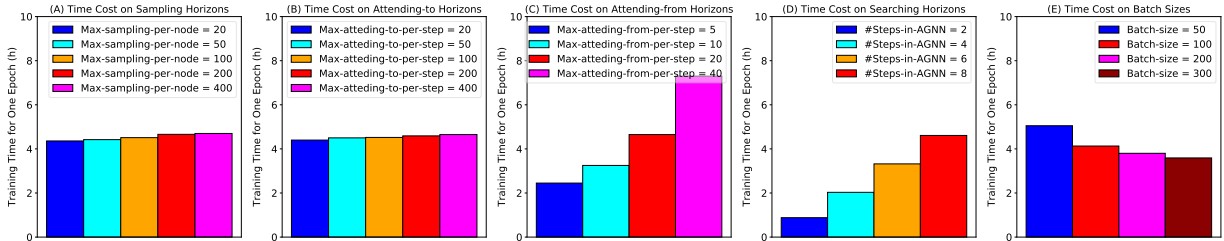

Figure 6: Analysis of time cost on WN18RR: (A)-(D) measure the one-epoch training time on different horizon settings corresponding to Figure 4(C)-(F); (E) measures on different batch sizes using horizon setting *Max-sampling-per-node*=20, *Max-attending-to-per-step*=20, *Max-attending-from-per-step*=20, and *#Steps-in-AGNN*=8. The charts on FB15K-237 can be found in the appendix.

from nodes per step is useful. That also explains why we call nodes in the attending-from horizon the *core nodes*, as they determine where subgraphs can be expanded and how attention will be propagated to affect the final probability distribution on the tail prediction. However, GPUs with a limited memory do not allow for a too large number of sampled or attended nodes especially for *Max-attending-from-per-step*. The detailed explanations can be found in attention strategies in Section 2 where the upper bound is controlled by $N_1N_2$ and $N_3$ (*Max-attending-from-per-step* corresponding to $N_1$, *Max-sampling-per-node* to $N_2$, and *Max-attending-to-per-step* to $N_3$). In $N_1N_2$, Section 3.2 suggests that we should sample more by a large $N_2$ but attend less by a small $N_1$. Figure 4(F) suggests that the propagation steps of AGNN should not go below four.

**Attention flow analysis.** If the flow-style attention really captures the way we reason about the world, its process should be conducted in a diverging-converging thinking pattern. Intuitively, first, for the diverging thinking phase, we search and collect ideas as much as we can; then, for the converging thinking phase, we try to concentrate our thoughts on one point. To check whether the attention flow has such a pattern, we measure the average entropy of attention distributions changing along steps and also the proportion of attention concentrated at the top-1,3,5 nodes. As we expect, attention is more focused at the final step and the beginning.

**Time cost analysis.** The time cost is affected not only by the scale of a dataset but also by the horizon setting. For each dataset, we list the training time for one epoch corresponding to our standard hyperparameter settings in the appendix. Note that there is always a trade-off between complexity and performance. We thus study whether we can reduce time cost a lot at the price of sacrificing a little performance. We plot the one-epoch training time in Figure 6(A)-(D), using the same settings as we do in the horizon analysis. We can see that *Max-attending-from-per-step* and *#Steps-in-AGNN* affect the training time significantly while *Max-sampling-per-node* and *Max-attending-to-per-step* affect very slightly. Therefore, we can use smaller *Max-sampling-per-node* and *Max-attending-to-per-step* in order to gain a larger batch size, making the computation more efficiency as shown in Figure 6(E).

**Visualization.** To further demonstrate the reasoning capability, we show visualization results of some pruned subgraphs on NELL995's test data for 12 separate tasks. We avoid using the training data in order to show generalization of the learned reasoning capability. We show the visualization results in Figure 1. See the appendix for detailed analysis and more visualization results.

**Discussion of the limitation.** Although DPMPN shows a promising way to harness the scalability on large-scale graph data, current GPU-based machine learning platforms, such as TensorFlow and PyTorch, seem not ready to fully leverage sparse tensor computation which acts as building blocks to support dynamical computation graphs which varies from one input to another. Extra overhead caused by extensive sparse operations will neutralize the benefits of exploiting sparsity.

## 5 RELATED WORK

**Knowledge graph reasoning.** Early work, including TransE (Bordes et al., 2013) and its analogues (Wang et al., 2014; Lin et al., 2015b; Ji et al., 2015), DistMult (Yang et al., 2015), ConvE (Dettmers et al., 2018) and ComplEx (Trouillon et al., 2016), focuses on learning embeddings of entities and relations. Some recent works of this line (Sun et al., 2018; Lacroix et al., 2018) achieve high accuracy. Another line aims to learn inference paths (Lao et al., 2011; Guu et al., 2015; Lin et al., 2015a; Toutanova et al., 2016; Chen et al., 2018; Lin et al., 2018) for knowledge graph reasoning, especially DeepPath (Xiong et al., 2017), MINERVA (Das et al., 2018), and M-Walk (Shen et al., 2018), which use RL to learn multi-hop relational paths. However, these approaches, based on policy gradients or Monte Carlo tree search, often suffer from low sample efficiency and sparse rewards, requiring a large number of rollouts and sophisticated reward function design. Other efforts include learning soft logical rules (Cohen, 2016; Yang et al., 2017) or compostional programs (Liang et al., 2016).

**Relational reasoning in Graph Neural Networks.** Relational reasoning is regarded as the key for combinatorial generalization, taking the form of entity- and relation-centric organization to reason about the composition structure of the world (Craik, 1952; Lake et al., 2017). A multitude of recent implementations (Battaglia et al., 2018) encode relational inductive biases into neural networks to exploit graph-structured representation, including graph convolution networks (GCNs) (Bruna et al., 2014; Henaff et al., 2015; Duvenaud et al., 2015; Kearnes et al., 2016; Defferrard et al., 2016; Niepert et al., 2016; Kipf & Welling, 2017; Bronstein et al., 2017) and graph neural networks (Scarselli et al., 2009; Li et al., 2016; Santoro et al., 2017; Battaglia et al., 2016; Gilmer et al., 2017). Variants of GNN architectures have been developed. Relation networks (Santoro et al., 2017) use a simple but effective neural module to model relational reasoning, and its recurrent versions (Santoro et al., 2018; Palm et al., 2018) do multi-step relational inference for long periods; Interaction networks (Battaglia et al., 2016) provide a general-purpose learnable physics engine, and two of its variants are visual interaction networks (Watters et al., 2017) and vertex attention interaction networks (Hoshen, 2017); Message passing neural networks (Gilmer et al., 2017) unify various GCNs and GNNs into a general message passing formalism by analogy to the one in graphical models.

**Attention mechanism on graphs.** Neighborhood attention operation can enhance GNNs' representation power (Velickovic et al., 2018; Hoshen, 2017; Wang et al., 2018; Kool, 2018). These approaches often use multi-head self-attention to focus on specific interactions with neighbors when aggregating messages, inspired by (Bahdanau et al., 2015; Lin et al., 2017; Vaswani et al., 2017). Most graph-based attention mechanisms attend over neighborhood in a single-hop fashion, and (Hoshen, 2017) claims that the multi-hop architecture does not help to model high-order interaction in experiments. However, a flow-style design of attention in (Xu et al., 2018b) shows a way to model long-range attention, stringing isolated attention operations by transition matrices.

## 6 CONCLUSION

We introduce *Dynamically Pruned Message Passing Networks* (DPMPN) and apply it to large-scale knowledge graph reasoning tasks. We propose to learn an input-dependent local subgraph which is progressively and selectively constructed to model a sequential reasoning process in knowledge graphs. We use graphical attention expression, a flow-style attention mechanism, to guide and prune the underlying message passing, making it scalable for large-scale graphs and also providing clear graphical interpretations. We also take the inspiration from the consciousness prior to develop a two-GNN framework to boost experimental performances.

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

# Appendix

## 7 PROOF

**Proposition.** *Given a graph $\mathcal{G}$ (undirected or directed in both directions), we assume the probability of the degree of an arbitrary node being less than or equal to $d$ is larger than $p$, i.e., $P(\deg(v) \leq d) > p, \forall v \in V$. Considering a sequence of consecutively expanding subgraphs $(G^0, G^1, \ldots, G^T)$, starting with $G^0 = \{v\}$, for all $t \geq 1$, we can ensure*

$$P\big(|V_{G^t}| \leq \frac{d(d-1)^t - 2}{d-2}\big) > p^{\frac{d(d-1)^{t-1}-2}{d-2}}. \tag{6}$$

*Proof.* We consider the extreme case of greedy consecutive expansion, where $G^t = G^{t-1} \cup \Delta G^t = G^{t-1} \cup \partial G^{t-1}$, since if this case satisfies the inequality, any case of consecutive expansion can also satisfy it. By definition, all the subgraphs $G^t$ are a connected graph. Here, we use $\Delta V^t$ to denote $V_{\Delta G^t}$ for short. In the extreme case, we can ensure that the newly added nodes $\Delta V^t$ at step $t$ only belong to the neighborhood of the last added nodes $\Delta V^{t-1}$. Since for $t \geq 2$ each node in $\Delta V^{t-1}$ already has at least one edge within $G^{t-1}$ due to the definition of connected graphs, we can have

$$P\big(|\Delta V^t| \leq |\Delta V^{t-1}|(d-1)\big) > p^{|\Delta V^{t-1}|}. \tag{7}$$

For $t = 1$, we have $P(|\Delta V^1| \leq d) > p$ and thus

$$P\big(|V_{G^1}| \leq 1 + d\big) > p. \tag{8}$$

For $t \geq 2$, based on $|V_{G^t}| = 1 + |\Delta V^1| + \ldots + |\Delta V^t|$, we obtain

$$P\big(|V_{G^t}| \leq 1 + d + d(d-1) + \ldots + d(d-1)^{t-1}\big) > p^{1 + d + d(d-1) + \ldots + d(d-1)^{t-2}}, \tag{9}$$

which is

$$P\big(|V_{G^t}| \leq \frac{d(d-1)^t - 2}{d-2}\big) > p^{\frac{d(d-1)^{t-1}-2}{d-2}}. \tag{10}$$

We can find that $t = 1$ also satisfies this inequality. $\qquad\square$

## 8    HYPERPARAMETER SETTINGS

Table 3: Our standard hyperparameter settings we use for each dataset plus their one-epoch training time. For experimental analysis, we only adjust one hyperparameter and keep the remaining fixed as the standard setting. For NELL995, the one-epoch training time means the average time cost of the 12 single-query-relation tasks.

| Hyperparameter | FB15K-237 | FB15K | WN18RR | WN18 | YAGO3-10 | NELL995 |
|---|---|---|---|---|---|---|
| *batch_size* | 80 | 80 | 100 | 100 | 100 | 10 |
| *n_dims_att* | 50 | 50 | 50 | 50 | 50 | 200 |
| *n_dims* | 100 | 100 | 100 | 100 | 100 | 200 |
| *max_sampling_per_step (in IGNN)* | 10000 | 10000 | 10000 | 10000 | 10000 | 10000 |
| *max_attending_from_per_step* | 20 | 20 | 20 | 20 | 20 | 100 |
| *max_sampling_per_node (in AGNN)* | 200 | 200 | 200 | 200 | 200 | 1000 |
| *max_attending_to_per_step* | 200 | 200 | 200 | 200 | 200 | 1000 |
| *n_steps_in_IGNN* | 2 | 1 | 2 | 1 | 1 | 1 |
| *n_steps_in_AGNN* | 6 | 6 | 8 | 8 | 6 | 5 |
| *learning_rate* | 0.001 | 0.001 | 0.001 | 0.001 | 0.0001 | 0.001 |
| *optimizer* | Adam | Adam | Adam | Adam | Adam | Adam |
| *grad_clipnorm* | 1 | 1 | 1 | 1 | 1 | 1 |
| *n_epochs* | 1 | 1 | 1 | 1 | 1 | 3 |
| One-epoch training time (h) | 25.7 | 63.7 | 4.3 | 8.5 | 185.0 | 0.12 |

The hyperparameters can be categorized into three groups:

- Normal hyperparameters, including *batch_size*, *n_dims_att*, *n_dims*, *learning_rate*, *grad_clipnorm*, and *n_epochs*. We set smaller dimensions, *n_dims_att*, for computation in the attention module, as it uses more edges than the message passing uses in AGNN, and also intuitively, it does not need to propagate high-dimensional messages but only compute scalar scores over a sampled neighborhood, in concert with the idea in the key-value mechanism (Bengio, 2017). We set $n\_epochs = 1$ in most cases, indicating that our model can be trained well by one epoch only due to its fast convergence.

- The hyperparameters in charge of the sampling-attending horizon, including *max_sampling_per_step* that controls the maximum number to sample edges per step in IGNN, and *max_sampling_per_node*, *max_attending_from_per_step* and *max_attending_to_per_step* that control the maximum number to sample neighbors of each selected node per step per input, the maximum number of selected nodes for attending-from per step per input, and the maximum number of selected nodes in a sampled neighborhood for attending-to per step per input in AGNN.

- The hyperparameters in charge of the searching horizon, including *n_steps_in_IGNN* representing the number of propagation steps to run standard message passing in IGNN, and *n_steps_in_AGNN* representing the number of propagation steps to run pruned message passing in AGNN.

Note that we tune these hyperparameters according to not only their performances but also the computation resources available to us. In some cases, to deal with a very large knowledge graph with limited resources, we need to make a trade-off between efficiency and effectiveness. For example, each of NELL995's single-query-relation tasks has a small training set, though still with a large graph, so we can reduce the batch size in favor of affording larger dimensions and a larger sampling-attending horizon without any concern for waiting too long to finish one epoch.

# 9 MORE EXPERIMENTAL RESULTS

Table 4: Comparison results on the FB15K and WN18 datasets. Results of [♠] are taken from (Nickel et al., 2016), [♣] from (Dettmers et al., 2018), [◇] from (Sun et al., 2018), [♡] from (Yang et al., 2017), and [✠] from (Lacroix et al., 2018). Our results take the form of "mean (std)".

| Metric (%) | FB15K | | | | WN18 | | | |
|---|---|---|---|---|---|---|---|---|
| | H@1 | H@3 | H@10 | MRR | H@1 | H@3 | H@10 | MRR |
| TransE [♠] | 29.7 | 57.8 | 74.9 | 46.3 | 11.3 | 88.8 | 94.3 | 49.5 |
| HolE [♠] | 40.2 | 61.3 | 73.9 | 52.4 | 93.0 | 94.5 | 94.9 | 93.8 |
| DistMult [♣] | 54.6 | 73.3 | 82.4 | 65.4 | 72.8 | 91.4 | 93.6 | 82.2 |
| ComplEx [♣] | 59.9 | 75.9 | 84.0 | 69.2 | 93.6 | 93.6 | 94.7 | 94.1 |
| ConvE [♣] | 55.8 | 72.3 | 83.1 | 65.7 | 93.5 | 94.6 | 95.6 | 94.3 |
| RotatE [◇] | **74.6** | **83.0** | 88.4 | 79.7 | **94.4** | **95.2** | 95.9 | 94.9 |
| ComplEx-N3 [✠] | - | - | **91** | **86** | - | - | 96 | 95 |
| NeuralLP [♡] | - | - | 83.7 | 76 | - | - | 94.5 | 94 |
| **DPMPN** | 72.6 (.4) | 78.4 (.4) | 83.4 (.5) | 76.4 (.4) | 91.6 (.8) | 93.6 (.4) | 94.9 (.4) | 92.8 (.6) |

Table 5: Comparison results on the YAGO3-10 dataset. Results of [♠] are taken from (Dettmers et al., 2018), [♣] from (Lacroix et al., 2018), and [✠] from (Lacroix et al., 2018).

| Metric (%) | YAGO3-10 | | | |
|---|---|---|---|---|
| | H@1 | H@3 | H@10 | MRR |
| DistMult [♠] | 24 | 38 | 54 | 34 |
| ComplEx [♠] | 26 | 40 | 55 | 36 |
| ConvE [♠] | 35 | 49 | 62 | 44 |
| ComplEx-N3 [✠] | - | - | **71** | **58** |
| **DPMPN** | **48.4** | **59.5** | 67.9 | 55.3 |

Table 6: Comparison results of MAP scores (%) on NELL995's single-query-relation KBC tasks. We take our baselines' results from (Shen et al., 2018). No reports found on the last two in the paper.

| Tasks | **NeuCFlow** | M-Walk | MINERVA | DeepPath | TransE | TransR |
|---|---|---|---|---|---|---|
| AthletePlaysForTeam | 83.9 (0.5) | **84.7 (1.3)** | 82.7 (0.8) | 72.1 (1.2) | 62.7 | 67.3 |
| AthletePlaysInLeague | 97.5 (0.1) | **97.8 (0.2)** | 95.2 (0.8) | 92.7 (5.3) | 77.3 | 91.2 |
| AthleteHomeStadium | **93.6 (0.1)** | 91.9 (0.1) | 92.8 (0.1) | 84.6 (0.8) | 71.8 | 72.2 |
| AthletePlaysSport | **98.6 (0.0)** | 98.3 (0.1) | **98.6 (0.1)** | 91.7 (4.1) | 87.6 | 96.3 |
| TeamPlayssport | **90.4 (0.4)** | 88.4 (1.8) | 87.5 (0.5) | 69.6 (6.7) | 76.1 | 81.4 |
| OrgHeadQuarteredInCity | 94.7 (0.3) | **95.0 (0.7)** | 94.5 (0.3) | 79.0 (0.0) | 62.0 | 65.7 |
| WorksFor | **86.8 (0.0)** | 84.2 (0.6) | 82.7 (0.5) | 69.9 (0.3) | 67.7 | 69.2 |
| PersonBornInLocation | **84.1 (0.5)** | 81.2 (0.0) | 78.2 (0.0) | 75.5 (0.5) | 71.2 | 81.2 |
| PersonLeadsOrg | 88.4 (0.1) | **88.8 (0.5)** | 83.0 (2.6) | 79.0 (1.0) | 75.1 | 77.2 |
| OrgHiredPerson | 84.7 (0.8) | **88.8 (0.6)** | 87.0 (0.3) | 73.8 (1.9) | 71.9 | 73.7 |
| AgentBelongsToOrg | **89.3 (1.2)** | - | - | - | - | - |
| TeamPlaysInLeague | **97.2 (0.3)** | - | - | - | - | - |

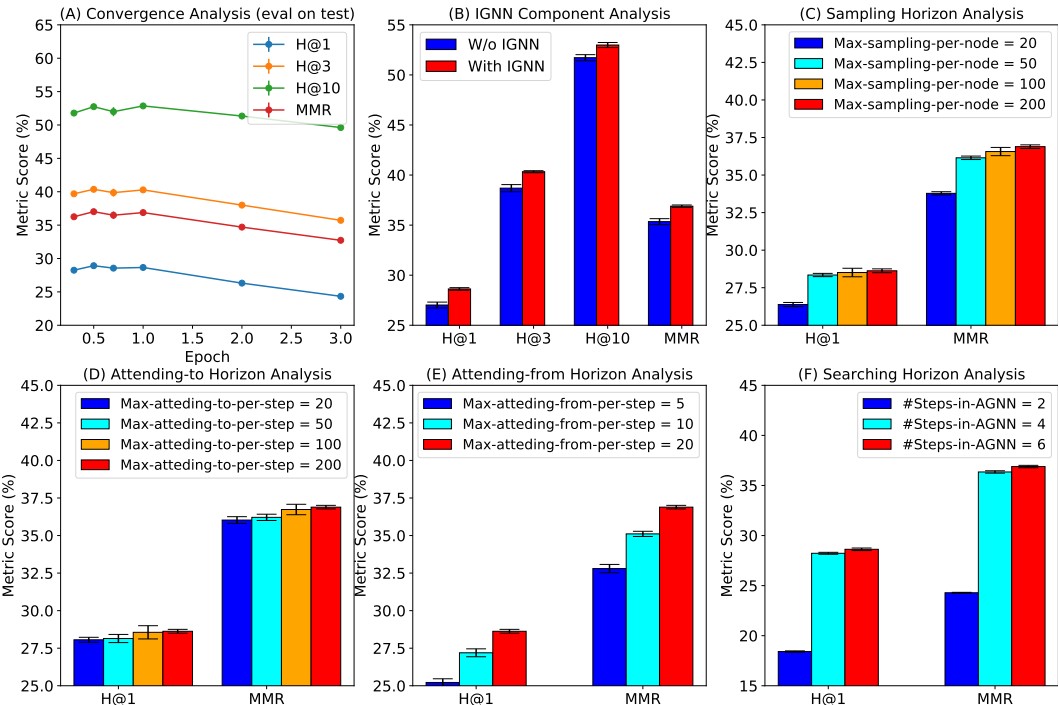

Figure 7: Experimental analysis on FB15K-237. (A) Convergence analysis: we pick six model snapshots at time points of 0.3, 0.5, 0.7, 1, 2, and 3 epochs during training and evaluate them on test; (B) IGNN component analysis: *w/o IGNN* uses zero step to run message passing, while *with IGNN* uses two steps; (C)-(F) Sampling, attending-to, attending-from and searching horizon analysis.

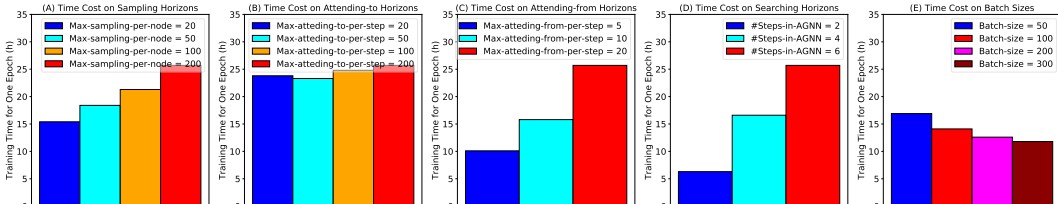

Figure 8: Analysis of time cost on FB15K-237: (A)-(D) measure the one-epoch training time on different horizon settings corresponding to Figure 7(C)-(F); (E) measures on different batch sizes using horizon setting *Max-sampled-edges-per-node*=20, *Max-seen-nodes-per-step*=20, *Max-attended-nodes-per-step*=20, and *#Steps-of-AGNN*=6.

## 10 MORE VISUALIZATION RESUTLS

### 10.1 CASE STUDY ON THE ATHLETEPLAYSFORTEAM TASK

In the case shown in Figure 9, the query is (*concept_personnorthamerica_michael_turner*, *concept:athleteplays-forteam*, ?) and a true answer is *concept_sportsteam_falcons*. From Figure 9, we can see our model learns that (*concept_personnorthamerica_michael_turner*, *concept:athletehomestadium*, *concept_stadiumoreventvenue_georgia_dome*) and (*concept_stadiumoreventvenue_georgia_dome*, *concept:teamhomestadium_inv*, *concept_sportsteam_falcons*) are two important facts to support the answer of *concept_sportsteam_falcons*. Besides, other facts, such as (*concept_athlete_joey_harrington*, *concept:athletehomestadium*, *concept_stadiumoreventvenue_georgia_dome*) and (*concept_athlete_joey_harrington*, *concept:athleteplaysforteam*, *concept_sportsteam_falcons*), provide a vivid example that a person or an athlete with *concept_stadiumoreventvenue_georgia_dome* as his or her home stadium might play for the team *concept_sportsteam_falcons*. We have such examples more than one, like *concept_athlete_roddy_white*'s and *concept_athlete_quarterback_matt_ryan*'s. The entity *con-*

*cept_sportsleague_nfl* cannot help us differentiate the true answer from other NFL teams, but it can at least exclude those non-NFL teams. In a word, our subgraph-structured representation can well capture the relational and compositional reasoning pattern.

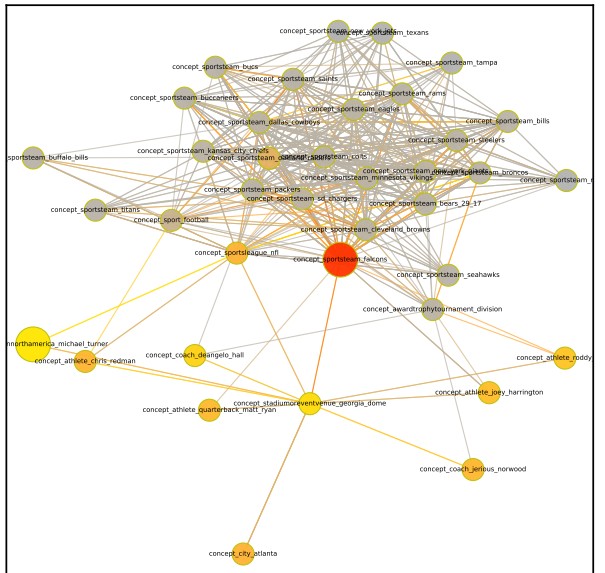 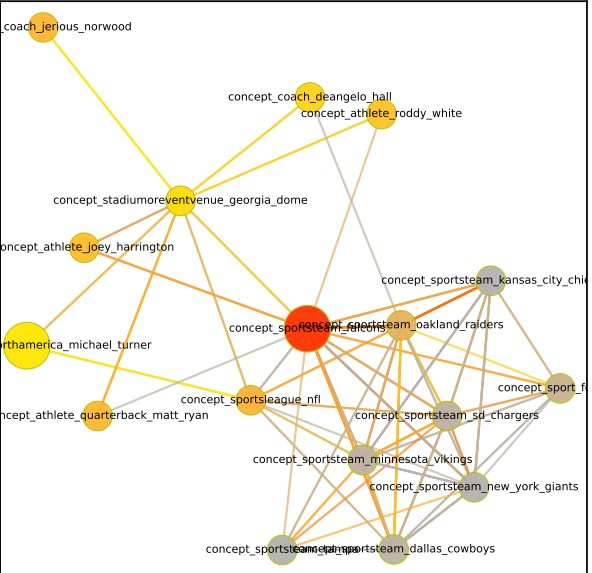

Figure 9: **AthletePlaysForTeam**. The head is *concept_personnorthamerica_michael_turner*, the query relation is *concept:athleteplaysforteam*, and the tail is *concept_sportsteam_falcons*. The left is a full subgraph derived with *max_attending_from_per_step*=20, and the right is a further pruned subgraph from the left based on attention. The big yellow node represents the head, and the big red node represents the tail. Color on the rest indicates attention scores over a $T$-step reasoning process, where grey means less attention, yellow means more attention gained during early steps, and red means gaining more attention when getting closer to the final step.

**For the AthletePlaysForTeam task**

```
Query: (concept_personnorthamerica_michael_turner, concept:athleteplaysforteam, concept_sportsteam_falcons)

Selected key edges:
concept_personnorthamerica_michael_turner, concept:agentbelongstoorganization, concept_sportsleague_nfl
concept_personnorthamerica_michael_turner, concept:athletehomestadium, concept_stadiumoreventvenue_georgia_dome
concept_sportsleague_nfl, concept:agentcompeteswithagent, concept_sportsleague_nfl
concept_sportsleague_nfl, concept:agentcompeteswithagent_inv, concept_sportsleague_nfl
concept_sportsleague_nfl, concept:teamplaysinleague_inv, concept_sportsteam_sd_chargers
concept_sportsleague_nfl, concept:leaguestadiums, concept_stadiumoreventvenue_georgia_dome
concept_sportsleague_nfl, concept:teamplaysinleague_inv, concept_sportsteam_falcons
concept_sportsleague_nfl, concept:agentbelongstoorganization_inv, concept_personnorthamerica_michael_turner
concept_stadiumoreventvenue_georgia_dome, concept:leaguestadiums_inv, concept_sportsleague_nfl
concept_stadiumoreventvenue_georgia_dome, concept:teamhomestadium_inv, concept_sportsteam_falcons
concept_stadiumoreventvenue_georgia_dome, concept:athletehomestadium_inv, concept_athlete_joey_harrington
concept_stadiumoreventvenue_georgia_dome, concept:athletehomestadium_inv, concept_athlete_roddy_white
concept_stadiumoreventvenue_georgia_dome, concept:athletehomestadium_inv, concept_coach_deangelo_hall
concept_stadiumoreventvenue_georgia_dome, concept:athletehomestadium_inv, concept_personnorthamerica_michael_turner
concept_sportsleague_nfl, concept:subpartoforganization_inv, concept_sportsteam_oakland_raiders
concept_sportsteam_sd_chargers, concept:teamplaysinleague, concept_sportsleague_nfl
concept_sportsteam_sd_chargers, concept:teamplaysagainstteam, concept_sportsteam_falcons
concept_sportsteam_sd_chargers, concept:teamplaysagainstteam_inv, concept_sportsteam_falcons
concept_sportsteam_sd_chargers, concept:teamplaysagainstteam, concept_sportsteam_oakland_raiders
concept_sportsteam_sd_chargers, concept:teamplaysagainstteam_inv, concept_sportsteam_oakland_raiders
concept_sportsteam_falcons, concept:teamplaysinleague, concept_sportsleague_nfl
concept_sportsteam_falcons, concept:teamplaysagainstteam, concept_sportsteam_sd_chargers
concept_sportsteam_falcons, concept:teamplaysagainstteam_inv, concept_sportsteam_sd_chargers
concept_sportsteam_falcons, concept:teamhomestadium, concept_stadiumoreventvenue_georgia_dome
concept_sportsteam_falcons, concept:teamplaysagainstteam, concept_sportsteam_oakland_raiders
concept_sportsteam_falcons, concept:teamplaysagainstteam_inv, concept_sportsteam_oakland_raiders
concept_sportsteam_falcons, concept:athleteledsportsteam_inv, concept_athlete_joey_harrington
concept_athlete_joey_harrington, concept:athletehomestadium, concept_stadiumoreventvenue_georgia_dome
concept_athlete_joey_harrington, concept:athleteledsportsteam, concept_sportsteam_falcons
concept_athlete_joey_harrington, concept:athleteplaysforteam, concept_sportsteam_falcons
concept_athlete_roddy_white, concept:athletehomestadium, concept_stadiumoreventvenue_georgia_dome
concept_athlete_roddy_white, concept:athleteplaysforteam, concept_sportsteam_falcons
concept_coach_deangelo_hall, concept:athletehomestadium, concept_stadiumoreventvenue_georgia_dome
```

```
concept_coach_deangelo_hall , concept : athleteplaysforteam , concept_sportsteam_oakland_raiders
concept_sportsleague_nfl , concept : teamplaysinleague_inv , concept_sportsteam_new_york_giants
concept_sportsteam_sd_chargers , concept : teamplaysagainstteam_inv , concept_sportsteam_new_york_giants
concept_sportsteam_falcons , concept : teamplaysagainstteam , concept_sportsteam_new_york_giants
concept_sportsteam_falcons , concept : teamplaysagainstteam_inv , concept_sportsteam_new_york_giants
concept_sportsteam_oakland_raiders , concept : teamplaysagainstteam_inv , concept_sportsteam_new_york_giants
concept_sportsteam_oakland_raiders , concept : teamplaysagainstteam , concept_sportsteam_sd_chargers
concept_sportsteam_oakland_raiders , concept : teamplaysagainstteam_inv , concept_sportsteam_sd_chargers
concept_sportsteam_oakland_raiders , concept : teamplaysagainstteam , concept_sportsteam_falcons
concept_sportsteam_oakland_raiders , concept : teamplaysagainstteam_inv , concept_sportsteam_falcons
concept_sportsteam_oakland_raiders , concept : agentcompeteswithagent , concept_sportsteam_oakland_raiders
concept_sportsteam_oakland_raiders , concept : agentcompeteswithagent_inv , concept_sportsteam_oakland_raiders
concept_sportsteam_new_york_giants , concept : teamplaysagainstteam , concept_sportsteam_sd_chargers
concept_sportsteam_new_york_giants , concept : teamplaysagainstteam , concept_sportsteam_falcons
concept_sportsteam_new_york_giants , concept : teamplaysagainstteam_inv , concept_sportsteam_falcons
concept_sportsteam_new_york_giants , concept : teamplaysagainstteam , concept_sportsteam_oakland_raiders
```

## 10.2  MORE RESULTS

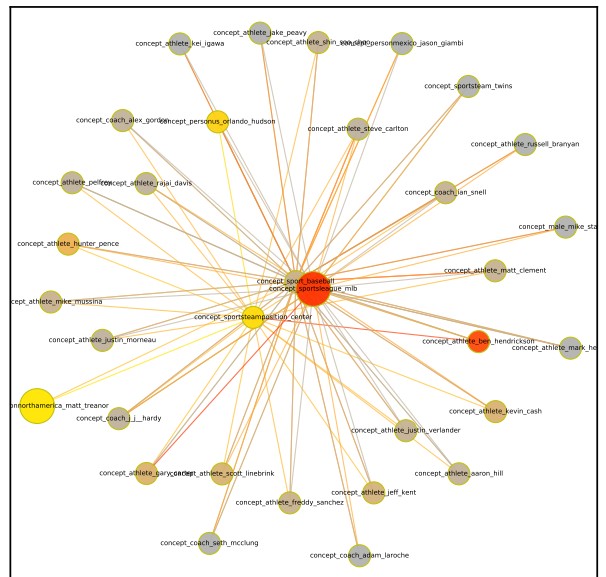 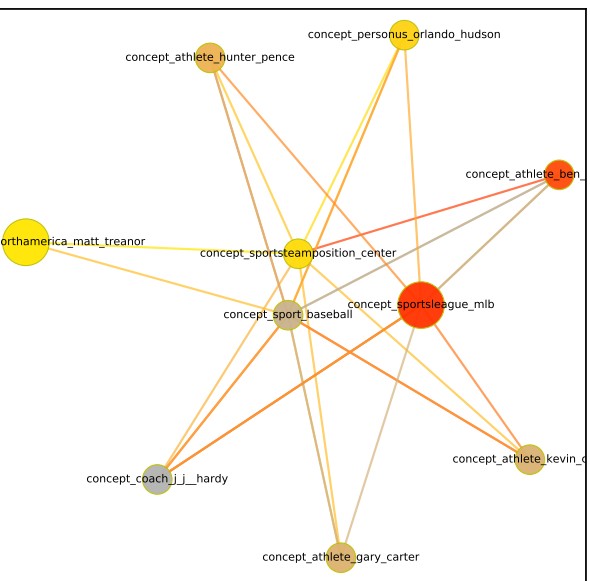

Figure 10: **AthletePlaysInLeague**. The head is *concept_personnorthamerica_matt_treanor*, the query relation is *concept:athleteplaysinleague*, and the tail is *concept_sportsleague_mlb*. The left is a full subgraph derived with *max_attending_from_per_step*=20, and the right is a further pruned subgraph from the left based on attention. The big yellow node represents the head, and the big red node represents the tail. Color on the rest indicates attention scores over a $T$-step reasoning process, where grey means less attention, yellow means more attention gained during early steps, and red means gaining more attention when getting closer to the final step.

**For the AthletePlaysInLeague task**

```
Query : ( concept_personnorthamerica_matt_treanor , concept : athleteplaysinleague , concept_sportsleague_mlb )

Selected key edges :
concept_personnorthamerica_matt_treanor , concept : athleteflyouttosportsteamposition , concept_sportsteamposition_center
concept_personnorthamerica_matt_treanor , concept : athleteplayssport , concept_sport_baseball
concept_sportsteamposition_center , concept : athleteflyouttosportsteamposition_inv , concept_personus_orlando_hudson
concept_sportsteamposition_center , concept : athleteflyouttosportsteamposition_inv , concept_athlete_ben_hendrickson
concept_sportsteamposition_center , concept : athleteflyouttosportsteamposition_inv , concept_coach_j_j__hardy
concept_sportsteamposition_center , concept : athleteflyouttosportsteamposition_inv , concept_athlete_hunter_pence
concept_sport_baseball , concept : athleteplayssport_inv , concept_personus_orlando_hudson
concept_sport_baseball , concept : athleteplayssport_inv , concept_athlete_ben_hendrickson
concept_sport_baseball , concept : athleteplayssport_inv , concept_coach_j_j__hardy
concept_sport_baseball , concept : athleteplayssport_inv , concept_athlete_hunter_pence
concept_personus_orlando_hudson , concept : athleteplaysinleague , concept_sportsleague_mlb
concept_personus_orlando_hudson , concept : athleteplayssport , concept_sport_baseball
concept_athlete_ben_hendrickson , concept : coachesinleague , concept_sportsleague_mlb
concept_athlete_ben_hendrickson , concept : athleteplayssport , concept_sport_baseball
```

```
concept_coach_j_j__hardy , concept:coachesinleague , concept_sportsleague_mlb
concept_coach_j_j__hardy , concept:athleteplaysinleague , concept_sportsleague_mlb
concept_coach_j_j__hardy , concept:athleteplayssport , concept_sport_baseball
concept_athlete_hunter_pence , concept:athleteplaysinleague , concept_sportsleague_mlb
concept_athlete_hunter_pence , concept:athleteplayssport , concept_sport_baseball
concept_sportsleague_mlb , concept:coachesinleague_inv , concept_athlete_ben_hendrickson
concept_sportsleague_mlb , concept:coachesinleague_inv , concept_coach_j_j__hardy
```

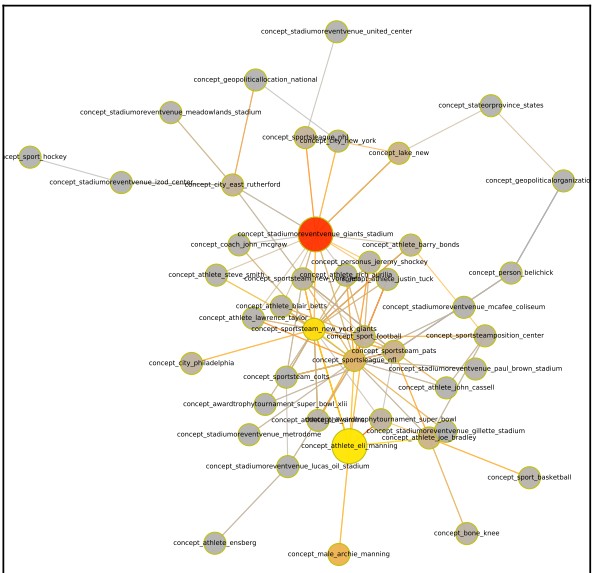 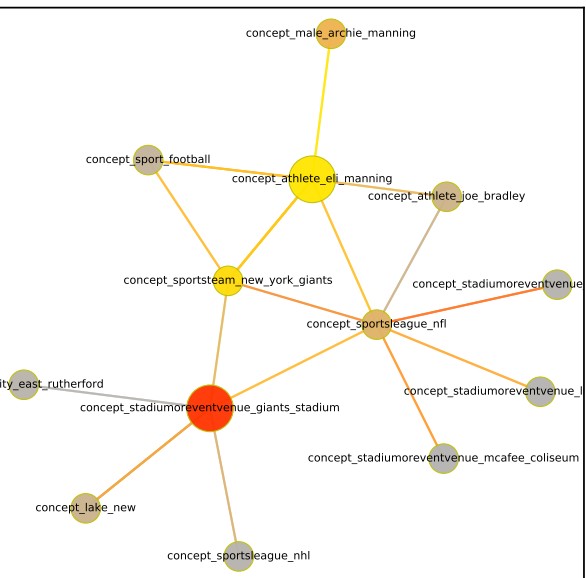

Figure 11: **AthleteHomeStadium**. The head is *concept_athlete_eli_manning*, the query relation is *concept:athletehomestadium*, and the tail is *concept_stadiumoreventvenue_giants_stadium*. The left is a full subgraph derived with *max_attending_from_per_step*=20, and the right is a further pruned subgraph from the left based on attention. The big yellow node represents the head, and the big red node represents the tail. Color on the rest indicates attention scores over a $T$-step reasoning process, where grey means less attention, yellow means more attention gained during early steps, and red means gaining more attention when getting closer to the final step.

**For the AthleteHomeStadium task**

```
Query: (concept_athlete_eli_manning , concept:athletehomestadium , concept_stadiumoreventvenue_giants_stadium)

Selected key edges:
concept_athlete_eli_manning , concept:personbelongstoorganization , concept_sportsteam_new_york_giants
concept_athlete_eli_manning , concept:athleteplaysforteam , concept_sportsteam_new_york_giants
concept_athlete_eli_manning , concept:athleteledsportsteam , concept_sportsteam_new_york_giants
concept_athlete_eli_manning , concept:athleteplaysinleague , concept_sportsleague_nfl
concept_athlete_eli_manning , concept:fatherofperson_inv , concept_male_archie_manning
concept_sportsteam_new_york_giants , concept:teamplaysinleague , concept_sportsleague_nfl
concept_sportsteam_new_york_giants , concept:teamhomestadium , concept_stadiumoreventvenue_giants_stadium
concept_sportsteam_new_york_giants , concept:personbelongstoorganization_inv , concept_athlete_eli_manning
concept_sportsteam_new_york_giants , concept:athleteplaysforteam_inv , concept_athlete_eli_manning
concept_sportsteam_new_york_giants , concept:athleteledsportsteam_inv , concept_athlete_eli_manning
concept_sportsleague_nfl , concept:teamplaysinleague_inv , concept_sportsteam_new_york_giants
concept_sportsleague_nfl , concept:agentcompeteswithagent , concept_sportsleague_nfl
concept_sportsleague_nfl , concept:agentcompeteswithagent_inv , concept_sportsleague_nfl
concept_sportsleague_nfl , concept:leaguestadiums , concept_stadiumoreventvenue_giants_stadium
concept_sportsleague_nfl , concept:athleteplaysinleague_inv , concept_athlete_eli_manning
concept_male_archie_manning , concept:fatherofperson , concept_athlete_eli_manning
concept_sportsleague_nfl , concept:leaguestadiums , concept_stadiumoreventvenue_paul_brown_stadium
concept_stadiumoreventvenue_giants_stadium , concept:teamhomestadium_inv , concept_sportsteam_new_york_giants
concept_stadiumoreventvenue_giants_stadium , concept:leaguestadiums_inv , concept_sportsleague_nfl
concept_stadiumoreventvenue_giants_stadium , concept:proxyfor_inv , concept_city_east_rutherford
concept_city_east_rutherford , concept:proxyfor , concept_stadiumoreventvenue_giants_stadium
concept_stadiumoreventvenue_paul_brown_stadium , concept:leaguestadiums_inv , concept_sportsleague_nfl
```

**For the AthletePlaysSport task**

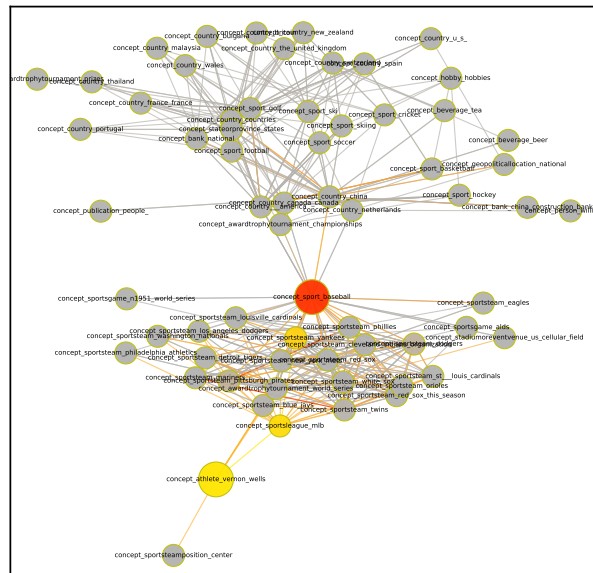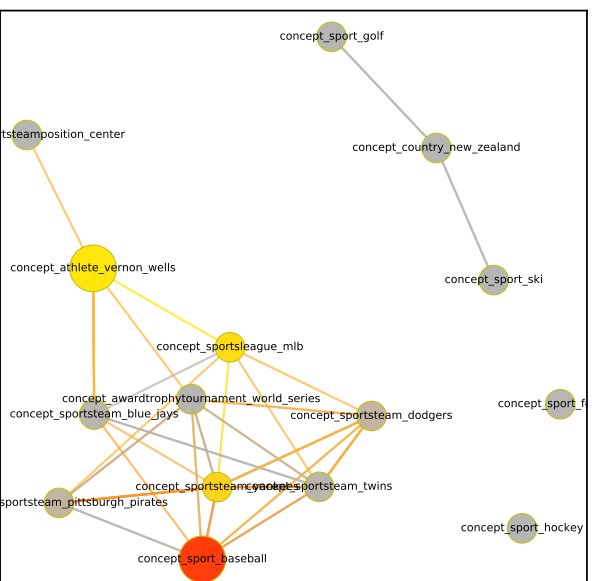

Figure 12: **AthletePlaysSport**. The head is *concept_athlete_vernon_wells*, the query relation is *concept:athleteplayssport*, and the tail is *concept_sport_baseball*. The left is a full subgraph derived with *max_attending_from_per_step*=20, and the right is a further pruned subgraph from the left based on attention. The big yellow node represents the head, and the big red node represents the tail. Color on the rest indicates attention scores over a $T$-step reasoning process, where grey means less attention, yellow means more attention gained during early steps, and red means gaining more attention when getting closer to the final step.

```
Query: (concept_athlete_vernon_wells, concept:athleteplayssport, concept_sport_baseball)

Selected key edges:
concept_athlete_vernon_wells, concept:athleteplaysinleague, concept_sportsleague_mlb
concept_athlete_vernon_wells, concept:coachwontrophy, concept_awardtrophytournament_world_series
concept_athlete_vernon_wells, concept:agentcollaborateswithagent_inv, concept_sportsteam_blue_jays
concept_athlete_vernon_wells, concept:personbelongstoorganization, concept_sportsteam_blue_jays
concept_athlete_vernon_wells, concept:athleteplaysforteam, concept_sportsteam_blue_jays
concept_athlete_vernon_wells, concept:athleteledsportsteam, concept_sportsteam_blue_jays
concept_sportsleague_mlb, concept:teamplaysinleague_inv, concept_sportsteam_dodgers
concept_sportsleague_mlb, concept:teamplaysinleague_inv, concept_sportsteam_yankees
concept_sportsleague_mlb, concept:teamplaysinleague_inv, concept_sportsteam_pittsburgh_pirates
concept_awardtrophytournament_world_series, concept:teamwontrophy_inv, concept_sportsteam_dodgers
concept_awardtrophytournament_world_series, concept:teamwontrophy_inv, concept_sportsteam_yankees
concept_awardtrophytournament_world_series, concept:awardtrophytournamentisthechampionshipgameofthenationalsport,
    concept_sport_baseball
concept_awardtrophytournament_world_series, concept:teamwontrophy_inv, concept_sportsteam_pittsburgh_pirates
concept_sportsteam_blue_jays, concept:teamplaysinleague, concept_sportsleague_mlb
concept_sportsteam_blue_jays, concept:teamplaysagainstteam, concept_sportsteam_yankees
concept_sportsteam_blue_jays, concept:teamplayssport, concept_sport_baseball
concept_sportsteam_dodgers, concept:teamplaysagainstteam, concept_sportsteam_yankees
concept_sportsteam_dodgers, concept:teamplaysagainstteam_inv, concept_sportsteam_yankees
concept_sportsteam_dodgers, concept:teamwontrophy, concept_awardtrophytournament_world_series
concept_sportsteam_dodgers, concept:teamplayssport, concept_sport_baseball
concept_sportsteam_yankees, concept:teamplaysagainstteam, concept_sportsteam_dodgers
concept_sportsteam_yankees, concept:teamplaysagainstteam_inv, concept_sportsteam_dodgers
concept_sportsteam_yankees, concept:teamwontrophy, concept_awardtrophytournament_world_series
concept_sportsteam_yankees, concept:teamplayssport, concept_sport_baseball
concept_sportsteam_yankees, concept:teamplaysagainstteam, concept_sportsteam_pittsburgh_pirates
concept_sportsteam_yankees, concept:teamplaysagainstteam_inv, concept_sportsteam_pittsburgh_pirates
concept_sport_baseball, concept:teamplayssport_inv, concept_sportsteam_dodgers
concept_sport_baseball, concept:teamplayssport_inv, concept_sportsteam_yankees
concept_sport_baseball, concept:awardtrophytournamentisthechampionshipgameofthenationalsport_inv,
    concept_awardtrophytournament_world_series
concept_sport_baseball, concept:teamplayssport_inv, concept_sportsteam_pittsburgh_pirates
concept_sportsteam_pittsburgh_pirates, concept:teamplaysagainstteam, concept_sportsteam_yankees
concept_sportsteam_pittsburgh_pirates, concept:teamplaysagainstteam_inv, concept_sportsteam_yankees
```

concept_sportsteam_pittsburgh_pirates , concept:teamwontrophy , concept_awardtrophytournament_world_series
concept_sportsteam_pittsburgh_pirates , concept:teamplayssport , concept_sport_baseball

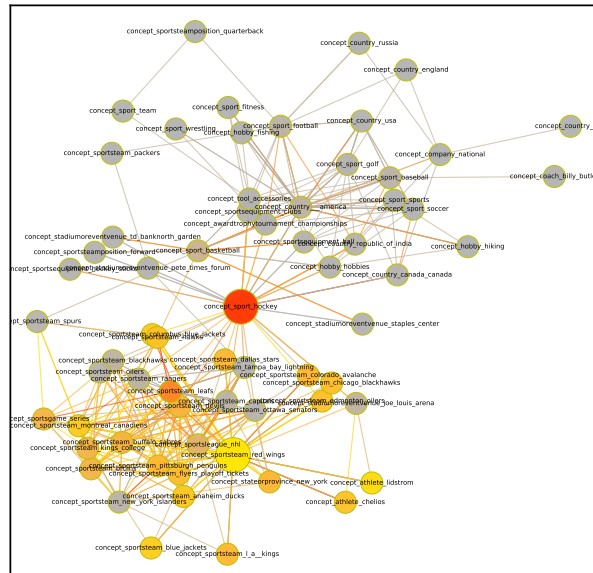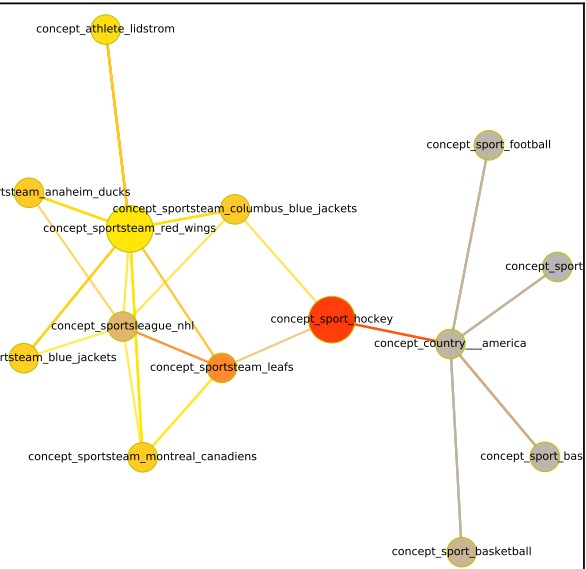

Figure 13: **TeamPlaysSport**. The head is *concept_sportsteam_red_wings*, the query relation is *concept:teamplayssport*, and the tail is *concept_sport_hockey*. The left is a full subgraph derived with *max_attending_from_per_step*=20, and the right is a further pruned subgraph from the left based on attention. The big yellow node represents the head, and the big red node represents the tail. Color on the rest indicates attention scores over a $T$-step reasoning process, where grey means less attention, yellow means more attention gained during early steps, and red means gaining more attention when getting closer to the final step.

**For the TeamPlaysSport task**

Query : ( concept_sportsteam_red_wings , concept:teamplayssport , concept_sport_hockey )

Selected key edges:
concept_sportsteam_red_wings , concept:teamplaysagainstteam , concept_sportsteam_montreal_canadiens
concept_sportsteam_red_wings , concept:teamplaysagainstteam_inv , concept_sportsteam_montreal_canadiens
concept_sportsteam_red_wings , concept:teamplaysagainstteam , concept_sportsteam_blue_jackets
concept_sportsteam_red_wings , concept:teamplaysagainstteam_inv , concept_sportsteam_blue_jackets
concept_sportsteam_red_wings , concept:worksfor_inv , concept_athlete_lidstrom
concept_sportsteam_red_wings , concept:organizationhiredperson , concept_athlete_lidstrom
concept_sportsteam_red_wings , concept:athleteplaysforteam_inv , concept_athlete_lidstrom
concept_sportsteam_red_wings , concept:athleteledsportsteam_inv , concept_athlete_lidstrom
concept_sportsteam_montreal_canadiens , concept:teamplaysagainstteam , concept_sportsteam_red_wings
concept_sportsteam_montreal_canadiens , concept:teamplaysagainstteam_inv , concept_sportsteam_red_wings
concept_sportsteam_montreal_canadiens , concept:teamplaysinleague , concept_sportsleague_nhl
concept_sportsteam_montreal_canadiens , concept:teamplaysagainstteam , concept_sportsteam_leafs
concept_sportsteam_montreal_canadiens , concept:teamplaysagainstteam_inv , concept_sportsteam_leafs
concept_sportsteam_blue_jackets , concept:teamplaysagainstteam , concept_sportsteam_red_wings
concept_sportsteam_blue_jackets , concept:teamplaysagainstteam_inv , concept_sportsteam_red_wings
concept_sportsteam_blue_jackets , concept:teamplaysinleague , concept_sportsleague_nhl
concept_athlete_lidstrom , concept:worksfor , concept_sportsteam_red_wings
concept_athlete_lidstrom , concept:organizationhiredperson_inv , concept_sportsteam_red_wings
concept_athlete_lidstrom , concept:athleteplaysforteam , concept_sportsteam_red_wings
concept_athlete_lidstrom , concept:athleteledsportsteam , concept_sportsteam_red_wings
concept_sportsteam_red_wings , concept:teamplaysinleague , concept_sportsleague_nhl
concept_sportsteam_red_wings , concept:teamplaysagainstteam , concept_sportsteam_leafs
concept_sportsteam_red_wings , concept:teamplaysagainstteam_inv , concept_sportsteam_leafs
concept_sportsleague_nhl , concept:agentcompeteswithagent , concept_sportsleague_nhl
concept_sportsleague_nhl , concept:agentcompeteswithagent_inv , concept_sportsleague_nhl
concept_sportsleague_nhl , concept:teamplaysinleague_inv , concept_sportsteam_leafs
concept_sportsteam_leafs , concept:teamplaysinleague , concept_sportsleague_nhl
concept_sportsteam_leafs , concept:teamplayssport , concept_sport_hockey

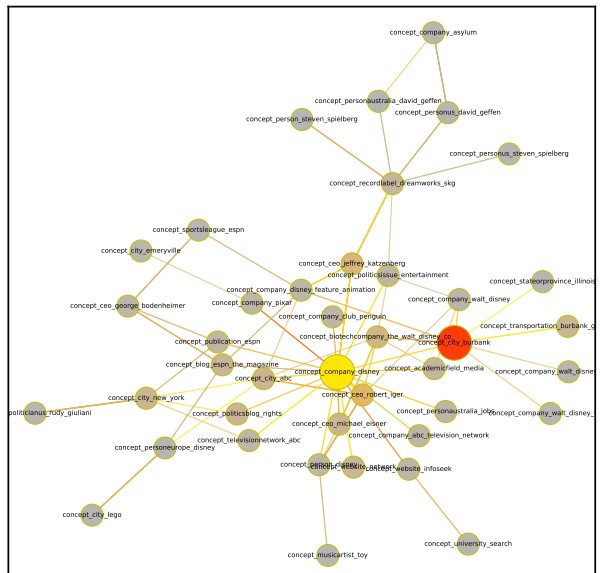 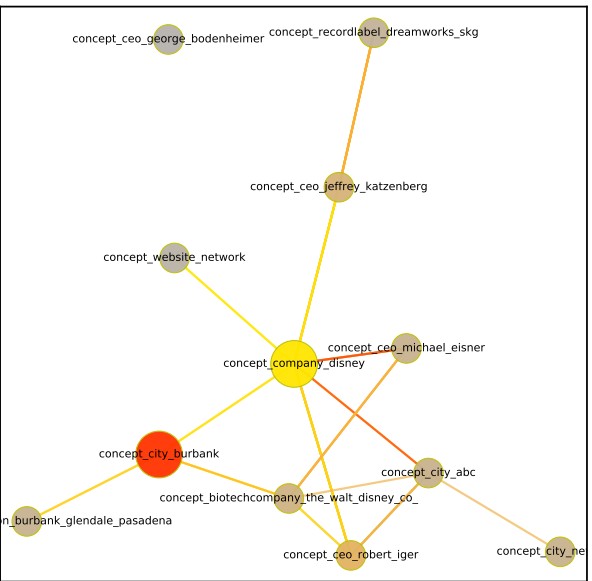

Figure 14: **OrganizationHeadQuarteredInCity**. The head is *concept_company_disney*, the query relation is *concept:organizationheadquaredincity*, and the tail is *concept_city_burbank*. The left is a full subgraph derived with *max_attending_from_per_step*=20, and the right is a further pruned subgraph from the left based on attention. The big yellow node represents the head, and the big red node represents the tail. Color on the rest indicates attention scores over a $T$-step reasoning process, where grey means less attention, yellow means more attention gained during early steps, and red means gaining more attention when getting closer to the final step.

**For the OrganizationHeadQuarteredInCity task**

Query: (concept_company_disney, concept:organizationheadquaredincity, concept_city_burbank)

Selected key edges:
concept_company_disney, concept:headquaredin, concept_city_burbank
concept_company_disney, concept:subpartoforganization_inv, concept_website_network
concept_company_disney, concept:worksfor_inv, concept_ceo_robert_iger
concept_company_disney, concept:proxyfor_inv, concept_ceo_robert_iger
concept_company_disney, concept:personleadsorganization_inv, concept_ceo_robert_iger
concept_company_disney, concept:ceoof_inv, concept_ceo_robert_iger
concept_company_disney, concept:personleadsorganization_inv, concept_ceo_jeffrey_katzenberg
concept_company_disney, concept:organizationhiredperson, concept_ceo_jeffrey_katzenberg
concept_company_disney, concept:organizationterminatedperson, concept_ceo_jeffrey_katzenberg
concept_city_burbank, concept:headquaredin_inv, concept_company_disney
concept_city_burbank, concept:headquaredin_inv, concept_biotechcompany_the_walt_disney_co_
concept_website_network, concept:subpartoforganization, concept_company_disney
concept_ceo_robert_iger, concept:worksfor, concept_company_disney
concept_ceo_robert_iger, concept:proxyfor, concept_company_disney
concept_ceo_robert_iger, concept:personleadsorganization, concept_company_disney
concept_ceo_robert_iger, concept:ceoof, concept_company_disney
concept_ceo_robert_iger, concept:topmemberoforganization, concept_biotechcompany_the_walt_disney_co_
concept_ceo_robert_iger, concept:organizationterminatedperson_inv, concept_biotechcompany_the_walt_disney_co_
concept_ceo_jeffrey_katzenberg, concept:personleadsorganization, concept_company_disney
concept_ceo_jeffrey_katzenberg, concept:organizationhiredperson_inv, concept_company_disney
concept_ceo_jeffrey_katzenberg, concept:organizationterminatedperson_inv, concept_company_disney
concept_ceo_jeffrey_katzenberg, concept:worksfor, concept_recordlabel_dreamworks_skg
concept_ceo_jeffrey_katzenberg, concept:topmemberoforganization, concept_recordlabel_dreamworks_skg
concept_ceo_jeffrey_katzenberg, concept:organizationterminatedperson_inv, concept_recordlabel_dreamworks_skg
concept_ceo_jeffrey_katzenberg, concept:ceoof, concept_recordlabel_dreamworks_skg
concept_biotechcompany_the_walt_disney_co_, concept:headquaredin, concept_city_burbank
concept_biotechcompany_the_walt_disney_co_, concept:organizationheadquaredincity, concept_city_burbank
concept_recordlabel_dreamworks_skg, concept:worksfor_inv, concept_ceo_jeffrey_katzenberg
concept_recordlabel_dreamworks_skg, concept:topmemberoforganization_inv, concept_ceo_jeffrey_katzenberg
concept_recordlabel_dreamworks_skg, concept:organizationterminatedperson, concept_ceo_jeffrey_katzenberg
concept_recordlabel_dreamworks_skg, concept:ceoof_inv, concept_ceo_jeffrey_katzenberg
concept_city_burbank, concept:airportincity_inv, concept_transportation_burbank_glendale_pasadena
concept_transportation_burbank_glendale_pasadena, concept:airportincity, concept_city_burbank

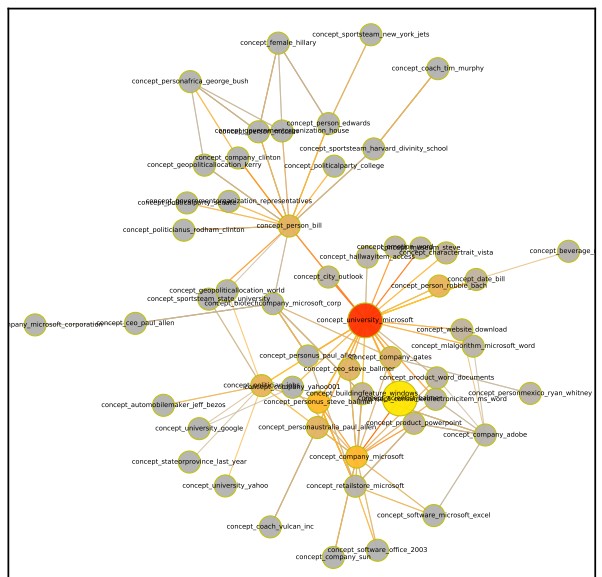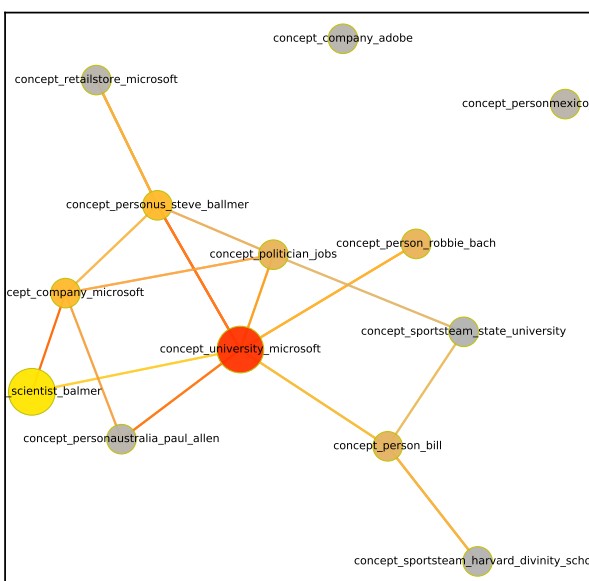

Figure 15: **WorksFor**. The head is *concept_scientist_balmer*, the query relation is *concept:worksfor*, and the tail is *concept_university_microsoft*. The left is a full subgraph derived with *max_attending_from_per_step*=20, and the right is a further pruned subgraph from the left based on attention. The big yellow node represents the head, and the big red node represents the tail. Color on the rest indicates attention scores over a $T$-step reasoning process, where grey means less attention, yellow means more attention gained during early steps, and red means gaining more attention when getting closer to the final step.

## For the WorksFor task

```
Query : ( concept_scientist_balmer , concept : worksfor , concept_university_microsoft )

Selected key edges :
concept_scientist_balmer , concept : topmemberoforganization , concept_company_microsoft
concept_scientist_balmer , concept : organizationterminatedperson_inv , concept_university_microsoft
concept_company_microsoft , concept : topmemberoforganization_inv , concept_personus_steve_ballmer
concept_company_microsoft , concept : topmemberoforganization_inv , concept_scientist_balmer
concept_university_microsoft , concept : agentcollaborateswithagent , concept_personus_steve_ballmer
concept_university_microsoft , concept : personleadsorganization_inv , concept_personus_steve_ballmer
concept_university_microsoft , concept : personleadsorganization_inv , concept_person_bill
concept_university_microsoft , concept : organizationterminatedperson , concept_scientist_balmer
concept_university_microsoft , concept : personleadsorganization_inv , concept_person_robbie_bach
concept_personus_steve_ballmer , concept : topmemberoforganization , concept_company_microsoft
concept_personus_steve_ballmer , concept : agentcollaborateswithagent_inv , concept_university_microsoft
concept_personus_steve_ballmer , concept : personleadsorganization , concept_university_microsoft
concept_personus_steve_ballmer , concept : worksfor , concept_university_microsoft
concept_personus_steve_ballmer , concept : proxyfor , concept_retailstore_microsoft
concept_personus_steve_ballmer , concept : subpartof , concept_retailstore_microsoft
concept_personus_steve_ballmer , concept : agentcontrols , concept_retailstore_microsoft
concept_person_bill , concept : personleadsorganization , concept_university_microsoft
concept_person_bill , concept : worksfor , concept_university_microsoft
concept_person_robbie_bach , concept : personleadsorganization , concept_university_microsoft
concept_person_robbie_bach , concept : worksfor , concept_university_microsoft
concept_retailstore_microsoft , concept : proxyfor_inv , concept_personus_steve_ballmer
concept_retailstore_microsoft , concept : subpartof_inv , concept_personus_steve_ballmer
concept_retailstore_microsoft , concept : agentcontrols_inv , concept_personus_steve_ballmer
```

## For the PersonBornInLocation task

```
Query : ( concept_person_mark001 , concept : personborninlocation , concept_county_york_city )

Selected key edges :
concept_person_mark001 , concept : persongraduatedfromuniversity , concept_university_college
concept_person_mark001 , concept : persongraduatedschool , concept_university_college
concept_person_mark001 , concept : persongraduatedfromuniversity , concept_university_state_university
concept_person_mark001 , concept : persongraduatedschool , concept_university_state_university
concept_person_mark001 , concept : personbornincity , concept_city_hampshire
```

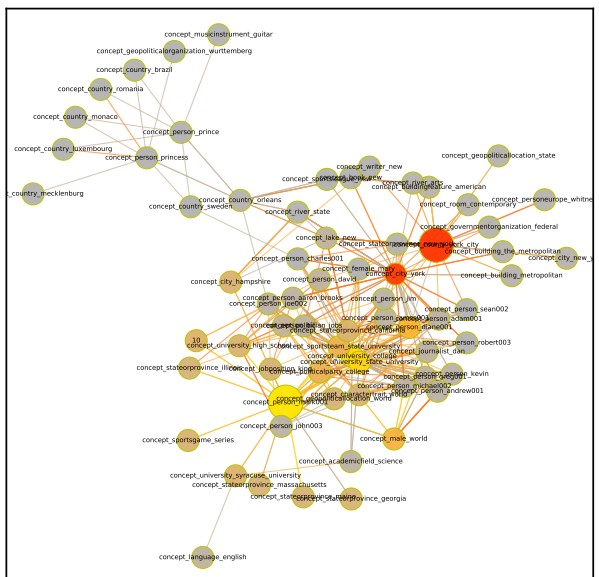 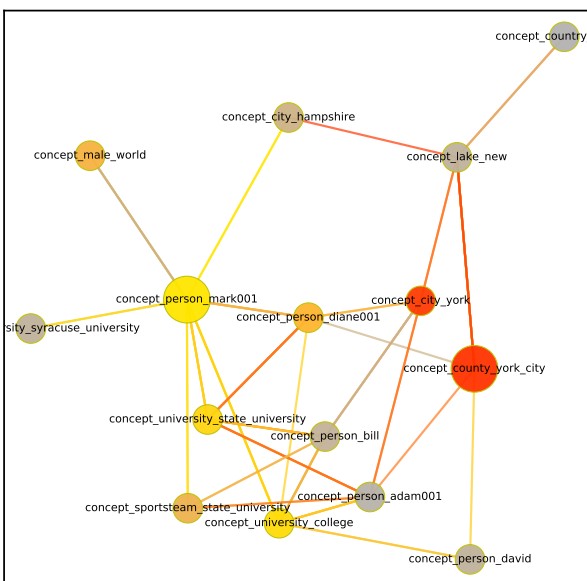

Figure 16: **PersonBornInLocation**. The head is *concept_person_mark001*, the query relation is *concept:personborninlocation*, and the tail is *concept_county_york_city*. The left is a full subgraph derived with *max_attending_from_per_step*=20, and the right is a further pruned subgraph from the left based on attention. The big yellow node represents the head, and the big red node represents the tail. Color on the rest indicates attention scores over a $T$-step reasoning process, where grey means less attention, yellow means more attention gained during early steps, and red means gaining more attention when getting closer to the final step.

```
concept_person_mark001 , concept:hasspouse , concept_person_diane001
concept_person_mark001 , concept:hasspouse_inv , concept_person_diane001
concept_university_college , concept:persongraduatedfromuniversity_inv , concept_person_mark001
concept_university_college , concept:persongraduatedschool_inv , concept_person_mark001
concept_university_college , concept:persongraduatedfromuniversity_inv , concept_person_bill
concept_university_college , concept:persongraduatedschool_inv , concept_person_bill
concept_university_state_university , concept:persongraduatedfromuniversity_inv , concept_person_mark001
concept_university_state_university , concept:persongraduatedschool_inv , concept_person_mark001
concept_university_state_university , concept:persongraduatedfromuniversity_inv , concept_person_bill
concept_university_state_university , concept:persongraduatedschool_inv , concept_person_bill
concept_city_hampshire , concept:personbornincity_inv , concept_person_mark001
concept_person_diane001 , concept:persongraduatedfromuniversity , concept_university_state_university
concept_person_diane001 , concept:persongraduatedschool , concept_university_state_university
concept_person_diane001 , concept:hasspouse , concept_person_mark001
concept_person_diane001 , concept:hasspouse_inv , concept_person_mark001
concept_person_diane001 , concept:personborninlocation , concept_county_york_city
concept_university_state_university , concept:persongraduatedfromuniversity_inv , concept_person_diane001
concept_university_state_university , concept:persongraduatedschool_inv , concept_person_diane001
concept_person_bill , concept:personbornincity , concept_city_york
concept_person_bill , concept:personborninlocation , concept_city_york
concept_person_bill , concept:persongraduatedfromuniversity , concept_university_college
concept_person_bill , concept:persongraduatedschool , concept_university_college
concept_person_bill , concept:persongraduatedfromuniversity , concept_university_state_university
concept_person_bill , concept:persongraduatedschool , concept_university_state_university
concept_city_york , concept:personbornincity_inv , concept_person_bill
concept_city_york , concept:personbornincity_inv , concept_person_diane001
concept_university_college , concept:persongraduatedfromuniversity_inv , concept_person_diane001
concept_person_diane001 , concept:personbornincity , concept_city_york
```

### For the PersonLeadsOrganization task

```
Query: (concept_journalist_bill_plante , concept:personleadsorganization , concept_company_cnn__pbs)

Selected key edges:
concept_journalist_bill_plante , concept:worksfor , concept_televisionnetwork_cbs
concept_journalist_bill_plante , concept:agentcollaborateswithagent_inv , concept_televisionnetwork_cbs
concept_televisionnetwork_cbs , concept:worksfor_inv , concept_journalist_walter_cronkite
```

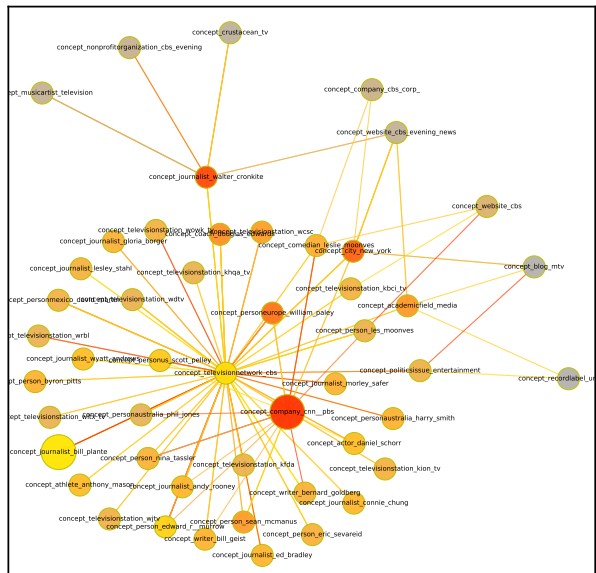 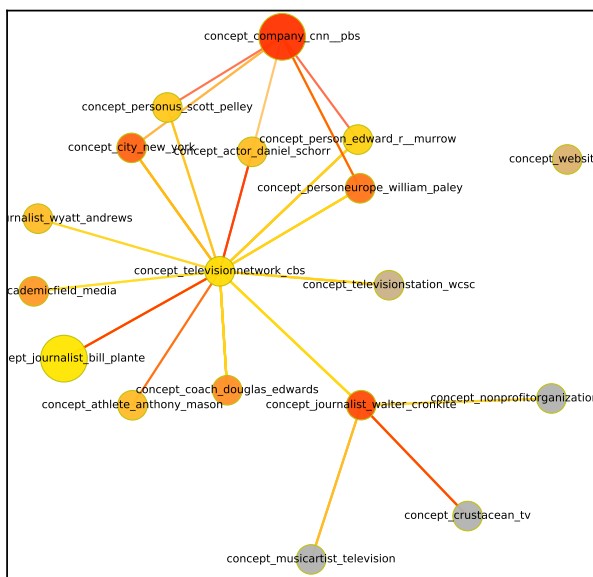

Figure 17: **PersonLeadsOrganization**. The head is *concept_journalist_bill_plante*, the query relation is *concept:organizationheadquarteredincity*, and the tail is *concept_company_cnn__pbs*. The left is a full subgraph derived with *max_attending_from_per_step*=20, and the right is a further pruned subgraph from the left based on attention. The big yellow node represents the head, and the big red node represents the tail. Color on the rest indicates attention scores over a $T$-step reasoning process, where grey means less attention, yellow means more attention gained during early steps, and red means gaining more attention when getting closer to the final step.

```
concept_televisionnetwork_cbs , concept:agentcollaborateswithagent , concept_journalist_walter_cronkite
concept_televisionnetwork_cbs , concept:worksfor_inv , concept_personus_scott_pelley
concept_televisionnetwork_cbs , concept:worksfor_inv , concept_actor_daniel_schorr
concept_televisionnetwork_cbs , concept:worksfor_inv , concept_person_edward_r__murrow
concept_televisionnetwork_cbs , concept:agentcollaborateswithagent , concept_person_edward_r__murrow
concept_televisionnetwork_cbs , concept:worksfor_inv , concept_journalist_bill_plante
concept_televisionnetwork_cbs , concept:agentcollaborateswithagent , concept_journalist_bill_plante
concept_journalist_walter_cronkite , concept:worksfor , concept_televisionnetwork_cbs
concept_journalist_walter_cronkite , concept:agentcollaborateswithagent_inv , concept_televisionnetwork_cbs
concept_journalist_walter_cronkite , concept:worksfor , concept_nonprofitorganization_cbs_evening
concept_personus_scott_pelley , concept:worksfor , concept_televisionnetwork_cbs
concept_personus_scott_pelley , concept:personleadsorganization , concept_televisionnetwork_cbs
concept_personus_scott_pelley , concept:personleadsorganization , concept_company_cnn__pbs
concept_actor_daniel_schorr , concept:worksfor , concept_televisionnetwork_cbs
concept_actor_daniel_schorr , concept:personleadsorganization , concept_televisionnetwork_cbs
concept_actor_daniel_schorr , concept:personleadsorganization , concept_company_cnn__pbs
concept_person_edward_r__murrow , concept:worksfor , concept_televisionnetwork_cbs
concept_person_edward_r__murrow , concept:agentcollaborateswithagent_inv , concept_televisionnetwork_cbs
concept_person_edward_r__murrow , concept:personleadsorganization , concept_televisionnetwork_cbs
concept_person_edward_r__murrow , concept:personleadsorganization , concept_company_cnn__pbs
concept_televisionnetwork_cbs , concept:organizationheadquateredincity , concept_city_new_york
concept_televisionnetwork_cbs , concept:headquateredin , concept_city_new_york
concept_televisionnetwork_cbs , concept:agentcollaborateswithagent , concept_personeurope_william_paley
concept_televisionnetwork_cbs , concept:topmemberoforganization_inv , concept_personeurope_william_paley
concept_company_cnn__pbs , concept:headquateredin , concept_city_new_york
concept_company_cnn__pbs , concept:personbelongstoorganization_inv , concept_personeurope_william_paley
concept_nonprofitorganization_cbs_evening , concept:worksfor_inv , concept_journalist_walter_cronkite
concept_city_new_york , concept:organizationheadquateredincity_inv , concept_televisionnetwork_cbs
concept_city_new_york , concept:headquateredin_inv , concept_televisionnetwork_cbs
concept_city_new_york , concept:headquateredin_inv , concept_company_cnn__pbs
concept_personeurope_william_paley , concept:agentcollaborateswithagent_inv , concept_televisionnetwork_cbs
concept_personeurope_william_paley , concept:topmemberoforganization , concept_televisionnetwork_cbs
concept_personeurope_william_paley , concept:personbelongstoorganization , concept_company_cnn__pbs
concept_personeurope_william_paley , concept:personleadsorganization , concept_company_cnn__pbs
```

**For the OrganizationHiredPerson task**

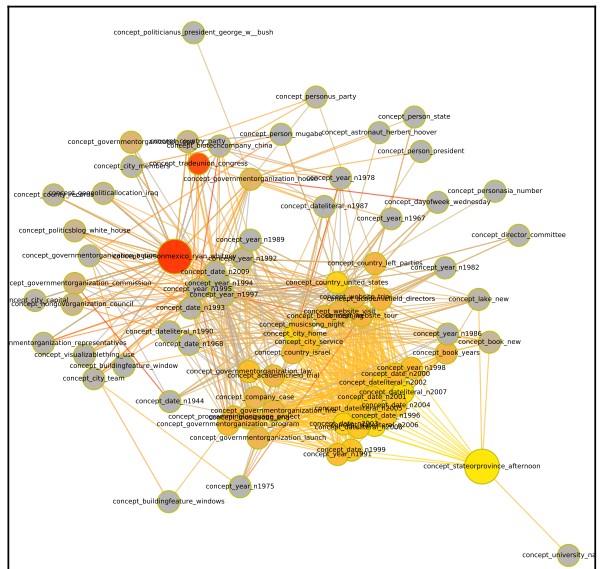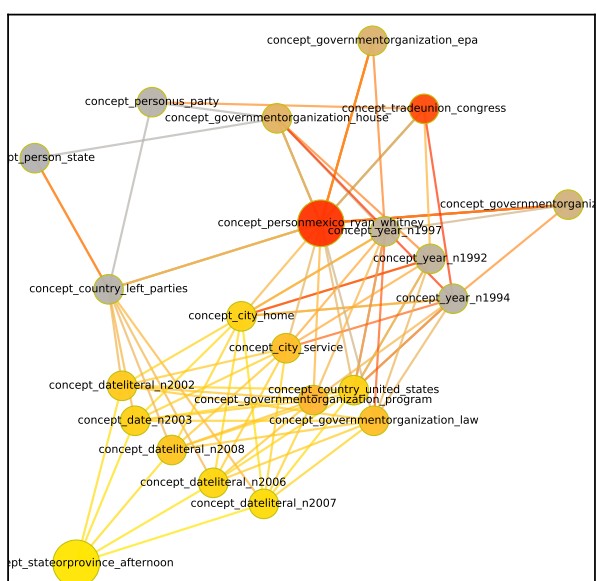

Figure 18: **OrganizationHiredPerson**. The head is *concept_stateorprovince_afternoon*, the query relation is *concept:organizationhiredperson*, and the tail is *concept_personmexico_ryan_whitney*. The left is a full subgraph derived with *max_attending_from_per_step*=20, and the right is a further pruned subgraph from the left based on attention. The big yellow node represents the head, and the big red node represents the tail. Color on the rest indicates attention scores over a $T$-step reasoning process, where grey means less attention, yellow means more attention gained during early steps, and red means gaining more attention when getting closer to the final step.

Query: ( concept_stateorprovince_afternoon , concept:organizationhiredperson , concept_personmexico_ryan_whitney )

Selected key edges:
concept_stateorprovince_afternoon , concept:atdate , concept_dateliteral_n2007
concept_stateorprovince_afternoon , concept:atdate , concept_date_n2003
concept_stateorprovince_afternoon , concept:atdate , concept_dateliteral_n2006
concept_dateliteral_n2007 , concept:atdate_inv , concept_country_united_states
concept_dateliteral_n2007 , concept:atdate_inv , concept_city_home
concept_dateliteral_n2007 , concept:atdate_inv , concept_city_service
concept_dateliteral_n2007 , concept:atdate_inv , concept_country_left_parties
concept_date_n2003 , concept:atdate_inv , concept_country_united_states
concept_date_n2003 , concept:atdate_inv , concept_city_home
concept_date_n2003 , concept:atdate_inv , concept_city_service
concept_date_n2003 , concept:atdate_inv , concept_country_left_parties
concept_dateliteral_n2006 , concept:atdate_inv , concept_country_united_states
concept_dateliteral_n2006 , concept:atdate_inv , concept_city_home
concept_dateliteral_n2006 , concept:atdate_inv , concept_city_service
concept_dateliteral_n2006 , concept:atdate_inv , concept_country_left_parties
concept_country_united_states , concept:atdate , concept_year_n1992
concept_country_united_states , concept:atdate , concept_year_n1997
concept_country_united_states , concept:organizationhiredperson , concept_personmexico_ryan_whitney
concept_city_home , concept:atdate , concept_year_n1992
concept_city_home , concept:atdate , concept_year_n1997
concept_city_home , concept:organizationhiredperson , concept_personmexico_ryan_whitney
concept_city_service , concept:atdate , concept_year_n1992
concept_city_service , concept:atdate , concept_year_n1997
concept_city_service , concept:organizationhiredperson , concept_personmexico_ryan_whitney
concept_country_left_parties , concept:worksfor_inv , concept_personmexico_ryan_whitney
concept_country_left_parties , concept:organizationhiredperson , concept_personmexico_ryan_whitney
concept_year_n1992 , concept:atdate_inv , concept_governmentorganization_house
concept_year_n1992 , concept:atdate_inv , concept_country_united_states
concept_year_n1992 , concept:atdate_inv , concept_city_home
concept_year_n1992 , concept:atdate_inv , concept_tradeunion_congress
concept_year_n1997 , concept:atdate_inv , concept_governmentorganization_house
concept_year_n1997 , concept:atdate_inv , concept_country_united_states
concept_year_n1997 , concept:atdate_inv , concept_city_home
concept_personmexico_ryan_whitney , concept:worksfor , concept_governmentorganization_house

```
concept_personmexico_ryan_whitney , concept:worksfor , concept_tradeunion_congress
concept_personmexico_ryan_whitney , concept:worksfor , concept_country_left_parties
concept_governmentorganization_house , concept:personbelongstoorganization_inv , concept_personus_party
concept_governmentorganization_house , concept:worksfor_inv , concept_personmexico_ryan_whitney
concept_governmentorganization_house , concept:organizationhiredperson , concept_personmexico_ryan_whitney
concept_tradeunion_congress , concept:organizationhiredperson , concept_personus_party
concept_tradeunion_congress , concept:worksfor_inv , concept_personmexico_ryan_whitney
concept_tradeunion_congress , concept:organizationhiredperson , concept_personmexico_ryan_whitney
concept_country_left_parties , concept:organizationhiredperson , concept_personus_party
```

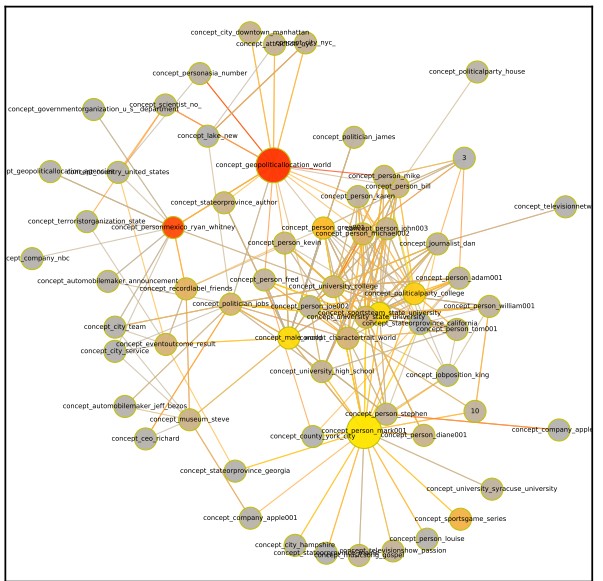 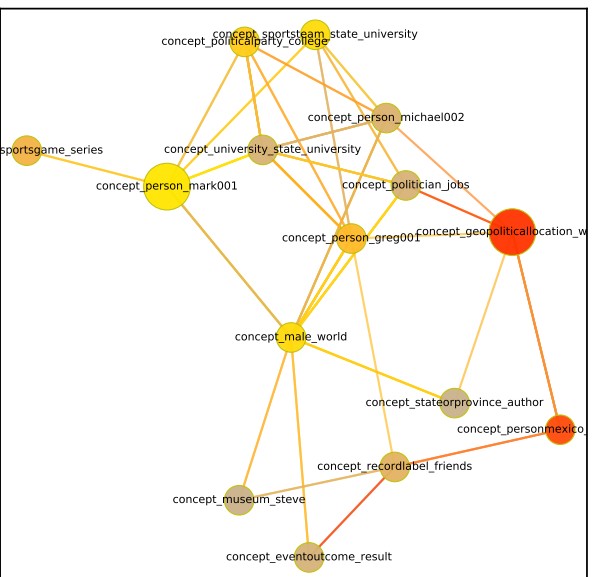

Figure 19: **AgentBelongsToOrganization**. The head is *concept_person_mark001*, the query relation is *concept:agentbelongstoorganization*, and the tail is *concept_geopoliticallocation_world*. The left is a full subgraph derived with *max_attending_from_per_step*=20, and the right is a further pruned subgraph from the left based on attention. The big yellow node represents the head, and the big red node represents the tail. Color on the rest indicates attention scores over a $T$-step reasoning process, where grey means less attention, yellow means more attention gained during early steps, and red means gaining more attention when getting closer to the final step.

**For the AgentBelongsToOrganization task**

```
Query: (concept_person_mark001 , concept:agentbelongstoorganization , concept_geopoliticallocation_world)

Selected key edges:
concept_person_mark001 , concept:personbelongstoorganization , concept_sportsteam_state_university
concept_person_mark001 , concept:agentcollaborateswithagent , concept_male_world
concept_person_mark001 , concept:agentcollaborateswithagent_inv , concept_male_world
concept_person_mark001 , concept:personbelongstoorganization , concept_politicalparty_college
concept_sportsteam_state_university , concept:personbelongstoorganization_inv , concept_politician_jobs
concept_sportsteam_state_university , concept:personbelongstoorganization_inv , concept_person_mark001
concept_sportsteam_state_university , concept:personbelongstoorganization_inv , concept_person_greg001
concept_sportsteam_state_university , concept:personbelongstoorganization_inv , concept_person_michael002
concept_male_world , concept:agentcollaborateswithagent , concept_politician_jobs
concept_male_world , concept:agentcollaborateswithagent_inv , concept_politician_jobs
concept_male_world , concept:agentcollaborateswithagent , concept_person_mark001
concept_male_world , concept:agentcollaborateswithagent_inv , concept_person_mark001
concept_male_world , concept:agentcollaborateswithagent , concept_person_greg001
concept_male_world , concept:agentcollaborateswithagent_inv , concept_person_greg001
concept_male_world , concept:agentcontrols , concept_person_greg001
concept_male_world , concept:agentcollaborateswithagent , concept_person_michael002
concept_male_world , concept:agentcollaborateswithagent_inv , concept_person_michael002
concept_politicalparty_college , concept:personbelongstoorganization_inv , concept_person_mark001
concept_politicalparty_college , concept:personbelongstoorganization_inv , concept_person_greg001
concept_politicalparty_college , concept:personbelongstoorganization_inv , concept_person_michael002
concept_politician_jobs , concept:personbelongstoorganization , concept_sportsteam_state_university
concept_politician_jobs , concept:agentcollaborateswithagent , concept_male_world
```

```
concept_politician_jobs , concept:agentcollaborateswithagent_inv , concept_male_world
concept_politician_jobs , concept:worksfor , concept_geopoliticallocation_world
concept_person_greg001 , concept:personbelongstoorganization , concept_sportsteam_state_university
concept_person_greg001 , concept:agentcollaborateswithagent , concept_male_world
concept_person_greg001 , concept:agentcollaborateswithagent_inv , concept_male_world
concept_person_greg001 , concept:agentcontrols_inv , concept_male_world
concept_person_greg001 , concept:agentbelongstoorganization , concept_geopoliticallocation_world
concept_person_greg001 , concept:personbelongstoorganization , concept_politicalparty_college
concept_person_greg001 , concept:agentbelongstoorganization , concept_recordlabel_friends
concept_person_michael002 , concept:personbelongstoorganization , concept_sportsteam_state_university
concept_person_michael002 , concept:agentcollaborateswithagent , concept_male_world
concept_person_michael002 , concept:agentcollaborateswithagent_inv , concept_male_world
concept_person_michael002 , concept:agentbelongstoorganization , concept_geopoliticallocation_world
concept_person_michael002 , concept:personbelongstoorganization , concept_politicalparty_college
concept_geopoliticallocation_world , concept:worksfor_inv , concept_personmexico_ryan_whitney
concept_geopoliticallocation_world , concept:organizationhiredperson , concept_personmexico_ryan_whitney
concept_geopoliticallocation_world , concept:worksfor_inv , concept_politician_jobs
concept_recordlabel_friends , concept:organizationhiredperson , concept_personmexico_ryan_whitney
concept_personmexico_ryan_whitney , concept:worksfor , concept_geopoliticallocation_world
concept_personmexico_ryan_whitney , concept:organizationhiredperson_inv , concept_geopoliticallocation_world
concept_personmexico_ryan_whitney , concept:organizationhiredperson_inv , concept_recordlabel_friends
```

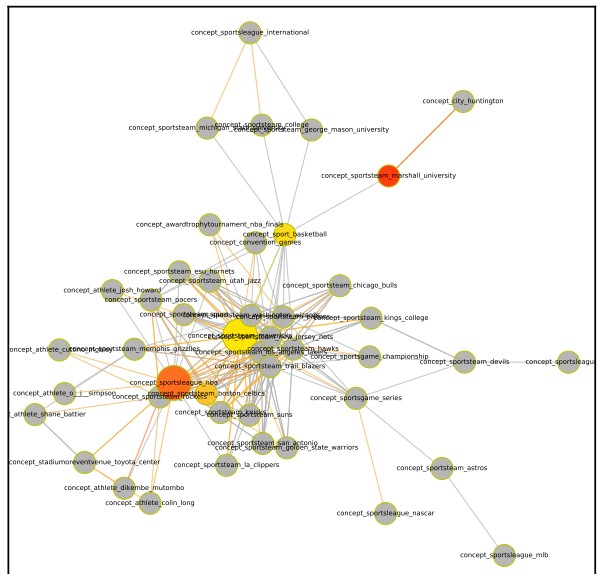 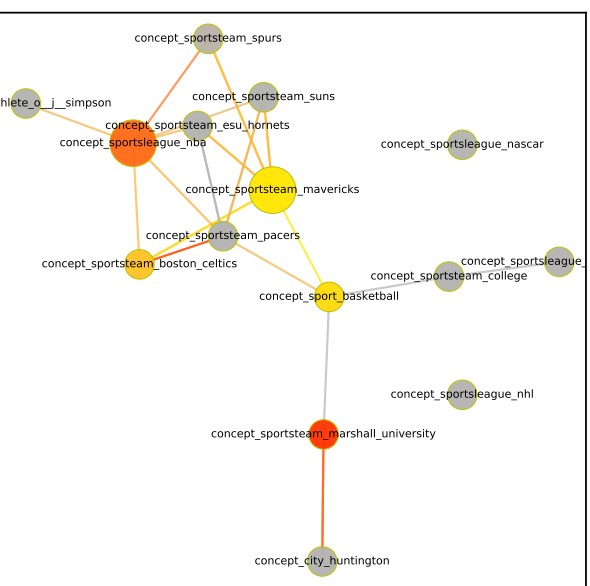

Figure 20: **TeamPlaysInLeague**. The head is *concept_sportsteam_mavericks*, the query relation is *concept:teamplaysinleague*, and the tail is *concept_sportsleague_nba*. The left is a full subgraph derived with *max_attending_from_per_step*=20, and the right is a further pruned subgraph from the left based on attention. The big yellow node represents the head, and the big red node represents the tail. Color on the rest indicates attention scores over a $T$-step reasoning process, where grey means less attention, yellow means more attention gained during early steps, and red means gaining more attention when getting closer to the final step.

**For the TeamPlaysInLeague task**

```
Query: (concept_sportsteam_mavericks , concept:teamplaysinleague , concept_sportsleague_nba)

Selected key edges:
concept_sportsteam_mavericks , concept:teamplayssport , concept_sport_basketball
concept_sportsteam_mavericks , concept:teamplaysagainstteam , concept_sportsteam_boston_celtics
concept_sportsteam_mavericks , concept:teamplaysagainstteam_inv , concept_sportsteam_boston_celtics
concept_sportsteam_mavericks , concept:teamplaysagainstteam , concept_sportsteam_spurs
concept_sportsteam_mavericks , concept:teamplaysagainstteam_inv , concept_sportsteam_spurs
concept_sport_basketball , concept:teamplayssport_inv , concept_sportsteam_college
concept_sport_basketball , concept:teamplayssport_inv , concept_sportsteam_marshall_university
concept_sportsteam_boston_celtics , concept:teamplaysinleague , concept_sportsleague_nba
concept_sportsteam_spurs , concept:teamplaysinleague , concept_sportsleague_nba
concept_sportsleague_nba , concept:agentcompeteswithagent , concept_sportsleague_nba
```

concept_sportsleague_nba , concept : agentcompeteswithagent_inv , concept_sportsleague_nba
concept_sportsteam_college , concept : teamplaysinleague , concept_sportsleague_international

