# OpenReview forum: "Dynamically Pruned Message Passing Networks for Large-scale Knowledge Graph Reasoning"
_ICLR.cc/2020/Conference — Accept (Poster)_

### Official Review · AnonReviewer3 · 2019-10-23
**Official Blind Review #3**

**Rating:** 6

**Review:**

The authors propose to use sampling methods in order to apply graph neural networks to large scale knowledge graphs for semantic reasoning. To this end induced subgraphs are constructed in a data-dependent way using an attention mechanism. This improves the efficiency and leads to interpretable results. The experiments show some improvements over path- and embedding-based methods.

The paper is partially difficult to read and not well structured (see minor comments below). Overall, I think that the proposed GNN architecture is an original and interesting approach for this specific application. The experimental evaluation presented in the Section 4 shows clear improvements. This, however, is not true for the results presented in the appendix (Table 4). I am missing a discussion of the limitations of the proposed approach. Moreover, a thorough discussion of the hyper-parameter selection and, if possible, theoretical justification would be highly desirable and could strengthen the paper.


Minor comments:

- Section 2 start with the paragraph 'Notation', but does not contain any other paragraph.

- The sampling strategy should not be introduced as part of the section 'problem formulation'.

- using standard terms from graph theory for well-known concepts (such as 'induced subgraph') would improve the readability

----------------------------
Update after the rebuttal: The authors have addressed several of my concerns and improved the manuscript. I have raised my score from "3: Weak Reject" to "6: Weak Accept".

**Experience Assessment:**

I have read many papers in this area.

**Review Assessment: Checking Correctness Of Derivations And Theory:**

N/A

**Review Assessment: Checking Correctness Of Experiments:**

I assessed the sensibility of the experiments.

**Review Assessment: Thoroughness In Paper Reading:**

I made a quick assessment of this paper.

---

> ### Author Response · Authors · 2019-11-11
> **Response to Review #3**
>
> Thank you for your valuable suggestions and pointing out my inappropriate word usage. We have replaced some terminology (e.g. using "induced subgraph" in our notation). We sincerely hope that our updated version could deliver a more reader-friendly presentation. We also appreciate that you recognized the originality of our work.
>
> ["The paper is partially difficult to read and not well structured (see minor comments below)"]
>
> - ["the paragraph 'Notation', but does not contain any other paragraph."]
>
> We feel sorry to make this paper difficult to read. To make it more concise and well structured, we rewrite some parts, including making other paragraphs in Section 2 have their own paragraph titles.
>
> - ["The sampling strategy should not be introduced as part of the section 'problem formulation'."]
>
> We realize this problem. We should not use the section title of "problem formulation" since we intend to explain and address the scale-up problem but not the typical "problem formulation" part in research papers. We made the correction in the revision by replacing the title with "Addressing the Scale-Up Problem".
>
> - ["well-known concepts (such as 'induced subgraph') would improve the readability"]
>
> We thank you for this suggestion. We used this term in many places in our updated version, which enables us to give a shorter and clearer description for the Notation part in Section 2.
>
> ["The experimental evaluation presented in the Section 4 shows clear improvements. This, however, is not true for the results presented in the appendix (Table 4)"]
>
> We feel that we should have given more background explanations on our datasets and why our model neither didn't perform well enough in WN18 as in other datasets, or didn't perform the best in FB15K. First, [1] noted  FB15K and WN18  are not challenging datasets because they contain many reversible triples. So, dataset FB15K-237 and WN18RR are created to serve as realistic KB completion datasets which represent a more challenging learning setting. However, this less-challenging property favors simple models compared to complex models such as path-based models. In our paper, we didn't just throw the results but presented them in the appendix. Though these results show our complex model may lose its advantage in easy datasets, our metric scores attained are still competitive and haven't become worse. In those challenging datasets, FB15K-237 and WN18RR, our model outperforms the best state-of-the-art significantly.
>
> ["I am missing a discussion of the limitations of the proposed approach."]
>
> This is a really good and useful suggestion. We have added the discussion of the limitations of our approach in the revision. We consider the main limitation could be:
> "Although DPMPN shows a promising way to harness the scalability on large-scale graph data, current GPU-based machine learning platforms, such as TensorFlow and PyTorch, seem not ready to fully leverage sparse tensor computation which acts as building blocks to support dynamical computation graphs which varies from one input to another. Extra overhead caused by extensive sparse operations will neutralize the benefits of exploiting sparsity."
>
> ["a thorough discussion of the hyper-parameter selection and, if possible, theoretical justification would be highly desirable"]
>
> Our hyper-parameters are listed in Section 8 in Appendix. You can see that main hyper-parameters are "max_attending_from_per_step", "max_sampling_per_node", "max_attending_to_per_step", "n_steps_in_IGNN" and "n_steps_in_AGNN", which define our attending-from, sampling, attending-to, and searching horizons. Then, we conduct a comprehensive ablation study to run many experiments on these hyperparameter selections and provide careful horizon analysis (see Figure 4(C)(D)(E)(F) and Figure 7(C)(D)(E)(F)). For theoretical justification, we only focus on the theoretical analysis to address the scale-up issue.
>
> [1] Kristina Toutanova and Danqi Chen. 2015. Observed Versus Latent Features for Knowledge Base and Text Inference. In Proceedings of the 3rd Workshop
>
> [2] Tim Dettmers, Pasquale Minervini, Pontus Stenetorp, and Sebastian Riedel. 2017. Convolutional 2D Knowledge Graph Embeddings. arXiv preprint abs/1707.01476. on Continuous Vector Space Models and their Compositionality. pages 57–66

---

### Official Review · AnonReviewer2 · 2019-10-23
**Official Blind Review #2**

**Rating:** 8

**Review:**

The paper presents a graph neural network model inspired by the consciousness prior of Bengio (2017) and implements it by means of two GNN models: the inattentive and the attentive GNN, respectively IGNN and AGNN.

Introduction outlines clearly what is the goal and why this is in principle an interesting avenue to investigate, however I found it could be more precise. In particular message passing techniques can be implemented very efficiently and are highly scalable. Section on Cognitive intuition of the consciousness prior doesn’t tell much beyond the re-stating of what it is, I am more interested in what it brings to the table and on what is the intuition behind its application to this domain.

One thing I would argue agains is that the input-dependent local subgraph requires access to the global graph, therefore going back to a conventional GNN. The argument is that the message passing can be constrained to reduce the computational burden. However, this can also be done by means of anchor graphs or other data structures, dynamic pooling and so fort. How do the authors compare to such choices?

Overall I like the “global conditioning“ by means of sampling to have better local representations at each node, an idea that while it can be cast as consciousness prior it is also related to neural processes and architectures alike.

The implementation section could be made a bit clearer. Currently, for instance, there are references to prediction tasks while I think it would be nicer to have it fully self contained. Also, the aggregation strategy seems to be that of Kipf et al., is it a constraint of the model or other strategies could be used as well?

The experiment section is clearly explained and results are interesting, showing the potential of the proposed approach. Also the thorough analysis of the model is very well done.
I wonder what is the performance when using no sampling at all, shouldn’t this be the reference?

The convergence analysis states that the model converges very fast, does it also translate to better results in case of small amount of training data?

**Experience Assessment:**

I do not know much about this area.

**Review Assessment: Checking Correctness Of Derivations And Theory:**

I did not assess the derivations or theory.

**Review Assessment: Checking Correctness Of Experiments:**

I carefully checked the experiments.

**Review Assessment: Thoroughness In Paper Reading:**

I read the paper at least twice and used my best judgement in assessing the paper.

---

> ### Author Response · Authors · 2019-11-11
> **Response to Review #2**
>
> Thank you for your careful and valuable reviews. We really appreciate your careful reading and providing useful suggestions to inspire our future research work.
>
> [About section on cognitive intuition of the consciousness prior]
> We consider our contribution to be another form of implementation of the consciousness prior to some extent by using GNNs instead of RNNs originally proposed in Bengio's paper. However, we feel that we might be still far from what the consciousness prior truly implies. I believe this work on knowledge graph reasoning is just the tip of an iceberg regarding what we should study in future to discover the true meaning of the consciousness prior. I am so glad that you are interested in our work and also this topic indeed.
>
> [About other means such as anchor graphs, dynamic pooling and so fort]
> Indeed, you give us a really good suggestion. We should explore them in our future work. However, in this work, especially in knowledge base completion tasks, almost all existing approaches are embedding-based or path-based. Therefore, to demonstrate the effectiveness of our work, we first consider the comparison between our model and those approaches.
>
> As you might see in other reviews, people may feel that our model architecture is so complicated. When applying Graph Neural Networks (GNNs) to knowledge graph reasoning, we didn't pay enough attention to Graph Convolution Networks (GCNs) but then realize that some GCNs might be easy to implement a pruned version to scale up. However, we worked out a pruned version of GNNs finally. Although it may sound more complex, we think it is more general and has more powerful capacity based on message passing mechanism.
>
> [About aggregation strategy and other strategies]
> We don't think the aggregation strategy will constrain our model. One of our contribution is to propose a graphical attention mechanism, which we call it attention flow. The point is that we only compute a transition matrix each step to transfer previous attention instead of generate new attention. In this way, the attention pipeline is separated from the message passing pipeline. Since the dynamically pruned subgraphs are attention-induced, our pruning procedure is free from the specific message passing mechanism including what the aggregation strategy we take. Therefore, for any form of Graph Neural Networks, we can just plug our flow-style attention module in to help do the pruning.
>
> [About sampling and attention]
> If we don't care about sampling at all, we should use a well-checked dataset which has a constrained node degree. Otherwise, the program will meet the out-of-memory error soon since a node with extremely high degree would bring an extremely large neighborhood and run out of resources immediately.
>
> Another problem is how attention is performed over a large candidate set. No matter how sparse we use attention mechanism to select the top-k, we have to do attention score computation over the full set first. Sampling can shrink the candidate set by performing it before computing attention scores.
>
> Based on the above arguments, it may show that sampling is no more than an engineering need. Indeed, we cannot provide theoretic justification to explain the necessity of sampling. However, we believe the consciousness prior proposed by Bengio. Sparsity might be the key, and sampling and attention will lead to this sparsity.
>
> [About the convergence]
> We should have explained the "fast" convergency in our experimental results more clearly. It is all about how we define an epoch. In standard training procedure, each input is independent from each other so that we cannot leverage other inputs when focusing on the current input. However, in our scenario, our operated graph data and the training data come from the same source, which means an input query <head, rel, tail> can appear as a graph edge used when training on other input queries. In this way, we exploit a triple <head, rel, tail> multiple times within one epoch. Indeed, our approach would cost more time each step, and that is because one training step here is equivalent to multiple steps in other models such as embedding models, and our one epoch amounts to multiple epochs in others.

---

### Official Review · AnonReviewer1 · 2019-10-23
**Official Blind Review #1**

**Rating:** 6

**Review:**

# Update after the rebuttal.
Thanks for reflecting some of my comments in the revision. The presentation seems to be improved, and the additional ablation study seems to address my concerns.

# Summary
This paper proposes a new neural network architecture for sequential reasoning task. The idea is to have two graph neural networks (GNNs), where one performs input-invariant global message passing, while the other performs input-dependent message passing locally. The input-dependent GNN employs a flow-style attention mechanism. The results on several knowledge completion datasets show that the proposed method outperforms the state-of-the-art methods.

# Originality
- The idea of learning an input-dependent subgraph using GNN seems new.
- The proposed way to reduce the complexity by restricting the attention horizon sounds interesting and seems necessary for scaling up.

# Quality
- The overall architecture looks like a fairly complicated combination of neural networks (two GNNs with attentive mechanism). However, it is not entirely clear how much each component contributes to the performance. The experiment only shows the benefit of having IGNN.
- The effect of the proposed complexity reduction technique is not studied in the experiment.
- The empirical results are hard to parse, as they contain too much dataset-specific results that are not clearly explain the paper.

# Clarity
- The paper is too dense with unnecessary details. For example, the introduction is too long (2.5 pages). The problem formulation contains too much details that deviate from the actual problem formulation. The details of each dataset (Table 1) and experimental setup can be moved to the appendix.
- Many figures in each experiment contain too small texts with lots of unexplained dataset-specific legends.

# Significance
- Although this paper proposes an interesting neural architecture for knowledge completion tasks, it is not clear how much each component contributes to the performance. Also, the empirical results could be presented in a better way to deliver clear conclusions.

**Experience Assessment:**

I do not know much about this area.

**Review Assessment: Checking Correctness Of Derivations And Theory:**

I did not assess the derivations or theory.

**Review Assessment: Checking Correctness Of Experiments:**

I assessed the sensibility of the experiments.

**Review Assessment: Thoroughness In Paper Reading:**

I made a quick assessment of this paper.

---

> ### Author Response · Authors · 2019-11-11
> **Response to Review #1**
>
> Thank you for your careful review and valuable comments. We have followed your suggestions to revise some parts in this paper and try to present it more concisely and reader-friendly. We sincerely hope that our efforts really address your concerns.
>
> # Originality
>
> We appreciate that you recognized the novelty and originality of this paper. We are glad to see that our work could interest you in addressing the scale-up issue in GNNs.
>
> # Quality
>
> ["it is not entirely clear how much each component contributes to the performance. The experiment only shows the benefit of having IGNN."]
>
> We agree that the ablation study on how much each component contributes to the performance is necessary. We did conduct experiments for component analysis to study whether IGNN is actually useful (see Figure 4(B) and Figure 7(B)). Since IGNN is input-agnostic, we cannot rely on its node representations only to predict a tail given an input query <head, rel, ?>. However, AGNN is input-dependent, which means it can be carried out to complete the knowledge base completion task without taking underlying node representations from IGNN. Therefore, we can arrange two sets of experiments: (1) AGNN + IGNN, and (2) AGNN-only. In this setting, we compare the first set which runs IGNN for two steps against the second one which totally shuts IGNN down.
>
> We feel sorry to cause your confusion of why "the experiment only shows the benefit of having IGNN". To make our experiments less confusing, we added more explanations in our revision about why we only arrange two sets without "IGNN-only".
>
> ["The effect of the proposed complexity reduction technique is not studied in the experiment."]
>
> Our proposed complexity reduction technique is described as the attending-sampling-attending procedure, where we define the attending-from horizon, the sampling horizon and the attending-to horizon in Section 3.2 to explain the procedure. Then, in experiments, we conduct a comprehensive ablation study on horizon analysis about how different complexity reduction hyperparameters affect the performance in terms of metric scores (see Figure 4(C)(D)(E) and Figure 7(C)(D)(E)), and also on time cost analysis between different hyperparameter selections (see Figure 6,8).
>
> ["The empirical results are hard to parse, as they contain too much dataset-specific results that are not clearly explain the paper."
>
> These six knowledge base datasets used in our paper, such as FB15K-237 and WN18RR, are used so widely in the domain of knowledge graph-related tasks. FB15K and FB15K-237 are from Freebase and WN18 and WN18RR are from WordNet. They are tested and used by many well-known published papers in this area, inicluding TransE [1], DistMult [2], ComplEx and DeepPath. Besides, we follow the same data processing and evaluation protocol as in many papers.
>
> # Clarity
>
> ["The paper is too dense with unnecessary details. For example, the introduction is too long (2.5 pages)"]
>
> We thank you for pointing out this. We have followed your suggestions to rewrite the introduction and provide a more concise version (within 1.5 pages). We hope that you can re-evaluate the revision to see whether it meets your expectation.
>
> ["The problem formulation contains too much details that deviate from the actual problem formulation."]
>
> We realize that we might misuse the section title of "Problem Formulation" which raised your confusion. Therefore, we replace it with "Addressing the Scale-Up Problem" which matches the section content more.
>
> ["Many figures in each experiment contain too small texts with lots of unexplained dataset-specific legends."]
>
> We feel sorry that the texts in our figures were too small to see. Therefore, we rearrange and enlarge our figures to make all texts clear to see. Those dataset-specific legends are all explained in the Appendix (see Section 8 Hyperparameter Settings). The reason that you felt "lots of unexplained dataset-specific legends" may be still due to the small texts not clear to see.
>
> # Significance
>
> ["the empirical results could be presented in a better way to deliver clear conclusions."]
>
> We conducted a very careful and thorough revision, attempting to address all your concerns. We readjusted figures to make it more clear and added analysis in experiments. We hope that this revision can better deliver our conclusions.
>
> [1] Antoine Bordes, Nicolas Usunier, Alberto Garc ́ıa-Dur ́an, Jason Weston, and Oksana Yakhnenko.Translating embeddings for modeling multi-relational data. InNIPS, 2013
> [2] Bishan Yang,  Wen tau Yih,  Xiaodong He,  Jianfeng Gao,  and Li Deng.   Embedding entities andrelations for learning and inference in knowledge bases.CoRR, abs/1412.6575, 2015
> [3] Th ́eo Trouillon, Johannes Welbl, Sebastian Riedel, ́Eric Gaussier, and Guillaume Bouchard.  Com-plex embeddings for simple link prediction. InICML, 2016
> [4] Wenhan Xiong, Thien Hoang, and William Yang Wang. Deeppath: A reinforcement learning methodfor knowledge graph reasoning. InEMNLP, 2017.

---

### Decision · Program_Chairs · 2019-12-19

**Decision:**

Accept (Poster)

**Comment:**

The paper presents a graph neural network model inspired by the consciousness prior of Bengio (2017) and implements it by means of two GNN models: the inattentive and the attentive GNN, respectively IGNN and AGNN. The reviewers think
- The idea of learning an input-dependent subgraph using GNN seems new.
- The proposed way to reduce the complexity by restricting the attention horizon sounds interesting.